# Mixture-of-Experts Can Surpass Dense LLMs Under Strictly Equal Resource

**Houyi Li**[1,2]
hyli22@m.fudan.edu.cn

**Ka Man Lo**[2]
kamanphoebe@gmail.com

**Shijie Xuyang**[1]
ysjxu24@m.fudan.edu.cn

**Ziqi Wang**[3]
wzq142857@mail.ustc.edu.cn

**Wenzhen Zheng**[2]
zhengwenzhen@amss.ac.cn

**Haocheng Zhang**[1]
hczhang25@m.fudan.edu.cn

**Zhao Li**[4]
lzjoey@gmail.com

**Shuigeng Zhou**[1*]
sgzhou@fudan.edu.cn

**Xiangyu Zhang**[2]
robert.zhang@stepfun.com

**Daxin Jiang**[2]
djiang@stepfun.com

[1]Fudan University    [2]StepFun
[3]University of Science and Technology of China    [4]Zhejiang University

## Abstract

Mixture-of-Experts (MoE) language models dramatically expand model capacity and achieve remarkable performance without increasing per-token compute. However, can MoEs *surpass* dense architectures under strictly equal resource constraints — that is, when the total parameter count, training compute, and data budget are identical? This question remains under-explored despite its significant practical value and potential. In this paper, we propose a novel perspective and methodological framework to study this question thoroughly. First, we comprehensively investigate the architecture of MoEs and achieve an optimal model design that maximizes the performance. Based on this, we subsequently find that an MoE model with activation rate in an optimal region is able to outperform its dense counterpart under the same total parameter, training compute and data resource. More importantly, this optimal region remains consistent across different model sizes. Although additional amount of data turns out to be a trade-off for enhanced performance, we show that this can be resolved via reusing data. We validate our findings through extensive experiments, training nearly 200 language models at 2B scale and over 50 at 7B scale, cumulatively processing 50 trillion tokens. All model checkpoints are publicly available[†].

## 1 Introduction

In recent years, Large Language Models (LLMs) built on the Transformer architecture (Vaswani et al., 2017) have achieved strong results on a range of NLP tasks (Radford et al., 2018; Achiam et al., 2023; Touvron et al., 2023a;b; Bai et al., 2023). Meanwhile, there has been growing interest in using Mixture-of-Experts (MoE) layers (Shazeer et al., 2017) to expand model capacity while keeping the training cost reasonable (Fedus et al., 2022; Zoph et al., 2022; Rajbhandari et al., 2022). Recent open-source initiatives have explored MoE-based LLMs (Dai et al., 2024; Jiang et al., 2024; Wei et al., 2024; Xue et al., 2024; DeepSeek-AI et al., 2024b), yet many widely adopted open-source models

---

*Corresponding author.    [†] https://huggingface.co/kamanphoebe/moe_surpass_dense.

such as LLaMA (Touvron et al., 2023a;b), DeepSeek's first-generation models (DeepSeek-AI et al., 2024a), and Qwen (Yang et al., 2024; Qwen et al., 2024), continue to utilize dense architectures, posing an open question of *whether MoE LLMs can outperform their dense counterparts*.

Current comparisons of MoE and dense LLMs often simplify the analysis to either a *data-centric* or a *compute-centric* perspective. The data-centric view, which keeps total training tokens constant for both MoE and dense models, praises MoE for its reduced activated parameter count per token (i.e., per-token compute cost) and potential for aggressive parameter scaling. For example, DeepSeek-MoE (Dai et al., 2024) reports a 16B-parameter MoE model (with 2.5B activated parameters) that achieves performance on par with a 7B dense model under the same data budget, thus suggesting a 2.5× "parameter-efficiency" advantage. The compute-centric view, which fixes total training compute, examines the impact of MoE sparsity (*i.e.*, the ratio of activated to total parameters) on performance. Under certain sparse configurations, total parameters can swell to nearly 100× those of a dense baseline, but at the cost of requiring more training data and managing high memory overhead.

However, neither perspectives fully addresses the complex interplay of critical resource constraints in large-scale model development: the finite nature of **training data volume** ($D$), **training compute** ($C$), and **model size** ($N$), which affects both memory and inference throughput. In particular, MoE models typically encounter bandwidth bottlenecks during inference, as all experts reside in GPU high-bandwidth memory and must be moved into shared memory, making parameter count a key runtime cost factor beyond FLOPs. These interdependencies complicate the conclusion of the absolute superiority of MoE or dense architectures. Intuitively, a dense model with equivalent total parameters should have an advantage by fully utilizing its capacity. This often leads studies to favor MoE for scaling model size rather than direct comparisons at the same parameter count, thus overlooking the real-world resource constraints in large-scale training and deployment.

In this work, we introduce a novel perspective aimed at providing a more convincing resolution to this debate by posing the following question:

> *Can Mixture-of-Experts surpass dense LLMs under equal total parameter, compute, and data constraints?*

To reach a definitive conclusion on this issue, we draw insights from a unified parameterization framework for model architecture (§ 3) and propose a three-step experimental methodology. *First*, we search for an optimized architecture design to ensure each model candidate achieves its (near-) optimal performance (§ 4). *Second*, we explore the optimal activation rate based on this optimized model architecture, with the total parameters $N$ and compute budget $C$ fixed (§ 5). *Third*, we present a data reuse strategy to address the additional data demand of MoE models, thereby equating data resource $D$ (§ 6). We also analyze the efficacy of this framework on downstream tasks (§ 7).

Our findings, derived from extensive experiments and systematic evaluation under the proposed strict $N/C/D$ parity with dense models, provide strong evidence that *MoE architectures with optimized backbones and activation rates can indeed achieve superior performance over dense models on both upstream and downstream tasks*. This implies that any observed performance gains can be attributed solely to architectural advantages, rather than disparities in parameter count or compute budget. Moreover, this challenges conventional wisdom and paves the way for resource-efficient yet powerful architectures in the next generation of large-scale NLP systems.

Our main contributions are: 1) We demonstrate, for the first time, that under fixed total parameters ($N$) and a fixed compute budget ($C$), an MoE LLM can surpass its dense counterpart with careful architecture design (see Fig. 1b, 2b). 2) Our experiments reveal the existence of a stable "optimal AR" region that consistently maximizes performance across varying $N$ (see Fig. 1, 2). 3) We introduce a practical data reuse strategy that offsets MoE's additional data needs, enabling robust gains over dense models without substantially increasing unique training data $D$ (see Fig. 2, 3).

## 2 RELATED WORK

### 2.1 MOE LANGUAGE MODELS

Building upon MoE (Shazeer et al., 2017), GShard (Lepikhin et al., 2021) facilitates model parallelism across devices for massive MoE models. With the advent of Transformers (Vaswani et al.,

2017), the integration of MoE into the Transformer framework has become a popular model architecture and achieved state-of-the-art performance. As an early attempt, Switch Transformer (Fedus et al., 2022) proposes top-1 gating to simplify MoE architecture and alleviate communication overhead. More recent Transformer-based MoE LLMs include (Xue et al., 2024; Jiang et al., 2024; Wu et al., 2024; Wei et al., 2024; DeepSeek-AI et al., 2024b). The MoE architecture is briefly reviewed in Appendix A.

## 2.2 Analyses of MoE Sparsity

Several studies investigated the impact of varying the number of MoE experts and adjusting granularity, both of which are factors related to sparsity. Through ablation studies, DeepSeekMoE (Dai et al., 2024) observes finer granularity results in improvement on overall model performance, and acquires a ratio between shared and routed experts that yields slightly better Pile loss. Zoph et al. (2022) summarized the results of several MoE works and indicated that the gain of increasing sparsity quickly diminishes when the number of experts is greater than 256 — a very sparse model.

Table 1: Notation.

| Symbol | Definition | Symbol | Definition |
|--------|-----------|--------|-----------|
| $D$ | Dataset size in tokens. | $S$ | Sequence length. |
| $M$ | Compute (w/o embedding) per token in FLOPs. | $H$ | Number of attention heads. |
| | | $D_\mathrm{m}$ | Model hidden dimension. |
| $C$ | Total training compute in FLOPs, *i.e.*, $M \cdot D$. | $D_\mathrm{ffn}$ | FFN hidden dimension. |
| | | $D_\mathrm{h}$ | Dimension of attention head. |
| $N$ | Number of non-vocabulary parameters. | $D_\mathrm{e}$ | Expert hidden dimension. |
| $N_\mathrm{a}$ | Number of activated parameters. | $D_\mathrm{se}$ | Shared expert hidden dimension. |
| $r_\mathrm{a}$ | Activation rate, *i.e.*, $N_\mathrm{a}/N$. | $E$ | Number of experts. |
| $L_\mathrm{e}$ | Number of MoE layers. | $K$ | Number of chosen experts. |
| $L_\mathrm{d}$ | Number of dense layers. | $\beta$ | Activated FFN-to-model ratio in MoE layers, *i.e.*, $(D_\mathrm{se} + KD_\mathrm{e})/D_\mathrm{m}$. |
| $L$ | Number of total layers, *i.e.*, $L_\mathrm{e} + L_\mathrm{d}$. | | |
| $\alpha$ | FFN expansion ratio, *i.e.*, $D_\mathrm{ffn}/D_\mathrm{m}$. | $\mu$ | Total FFN-to-model ratio in MoE layers, *i.e.*, $(D_\mathrm{se} + ED_\mathrm{e})/D_\mathrm{m}$. |
| $\zeta$ | Model aspect ratio, *i.e.*, $D_\mathrm{m}/L$. | | |
| $\gamma$ | Sequence-to-width ratio, *i.e.*, $S/D_\mathrm{m}$. | | |

From a methodological perspective, our comparisons are conducted in a sufficiently trained regime (with $D/N \geq 20$ for key models) and under a fixed-$N$ setting motivated by deployment memory constraints, which differs from scaling-law sweeps that often rely on undertrained large models at limited compute. We provide a more detailed discussion in Appendix B.

Concurrently with our work, Ludziejewski et al. (2025) found that a sufficiently large MoEs trained with more tokens outperforms a dense model with the same total parameters. We further show MoE superiority even at smaller sizes and address the additional data demand via reuse. Abnar et al. (2025) studied the scaling law for optimal MoE sparsity. However, their models (up to $N = 30B$) were trained with $C = 1e20$, a much smaller budget compared to the approximately $9\times$ and $30\times$ compute we used for our 2B and 7B models to ensure an adequate $D/N$. This likely resulted in undertrained models, potentially affecting their conclusions. Moreover, our study first optimizes the MoE architecture to isolate the effect of different activation rates on performance. Detailed differences between our work and this previous study are discussed in Appendix B.

## 3 Experimental Methodology

We begin by introducing a unified parameterization framework for model architecture, establishing a solid foundation. Then, we derive key insights from this parameterization, which informs our comprehensive three-step experimental methodology. Finally, we detail the experimental setup used consistently across all subsequent experiments. Our notation is summarized in Table 1.

## 3.1 ARCHITECTURE PARAMETERIZATION

To enable a comprehensive and general comparison of dense and MoE-based LLM architectures under realistic deployment constraints, we introduce a unified parameterization framework that explicitly accounts for both model parameters and per-token compute cost.

**Dense model parameterization.**  For a dense model, we approximate the number of non-embedding parameters $N$ and the per-token forward-pass computation cost $M$ as follows:

$$N \approx (4 + 3\alpha)D_{\mathrm{m}}^2 L = (4 + 3\alpha)\zeta^2 L^3, \tag{1}$$

$$M \approx 2N + 4D_{\mathrm{m}}SL = 2N + 4\zeta^2\gamma L^3 \tag{2}$$

$$= 2N(1 + {}^{2\gamma}/_{(4+3\alpha)}), \tag{3}$$

where $\alpha = {}^{D_{\mathrm{ffn}}}/_{D_{\mathrm{m}}}, \gamma = {}^{S}/_{D_{\mathrm{m}}}$, and $\zeta = {}^{D_{\mathrm{m}}}/_{L}$. Here, we omit the LayerNorm parameter count as it is negligible. Inference cost or training cost can be approximated by $C \approx M \times D$ or $C \approx 3M \times D$, respectively (based on the standard empirical observation that training typically takes about three times the forward pass for backward computation).

**MoE model parameterization.**  In many real-world settings, only a subset of Transformer layers are replaced with MoE layers. The approximations for total non-vocabulary parameters $N$, activated parameters $N_{\mathrm{a}}$, and per-token computation cost $M$ can be expressed as

$$N \approx (4 + 3\mu)D_{\mathrm{m}}^2 L_{\mathrm{e}} + (4 + 3\alpha)D_{\mathrm{m}}^2 L_{\mathrm{d}}, \tag{4}$$

$$N_{\mathrm{a}} \approx (4 + 3\beta)D_{\mathrm{m}}^2 L_{\mathrm{e}} + (4 + 3\alpha)D_{\mathrm{m}}^2 L_{\mathrm{d}}, \tag{5}$$

$$M \approx N_{\mathrm{a}} + 4D_{\mathrm{m}}SL = 2r_{\mathrm{a}}N + 4\zeta^2\gamma L^3, \tag{6}$$

where $\mu = {}^{(D_{\mathrm{se}} + ED_{\mathrm{e}})}/_{D_{\mathrm{m}}}$ and $\beta = {}^{(D_{\mathrm{se}} + KD_{\mathrm{e}})}/_{D_{\mathrm{m}}}$. We again omit the parameters and FLOPs of the gating network (router) as they are comparatively small, and the *activation rate* (AR) is denoted as $r_{\mathrm{a}}$. For the simple and common case where *all* $L$ layers are MoE layers (*i.e.*, $L_{\mathrm{d}} = 0$), we have

$$r_{\mathrm{a}} = N_{\mathrm{a}}/N = {}^{(4+3\beta)}/_{(4+3\mu)}, \tag{7}$$

$$M \approx 2r_{\mathrm{a}}N + 4\zeta^2\gamma L^3 \tag{8}$$

$$= 2r_{\mathrm{a}}N(1 + {}^{2\gamma}/_{(4+3\beta)}). \tag{9}$$

## 3.2 KEY OBSERVATIONS AND METHODOLOGICAL CONSIDERATIONS

Based on the above parameterization, we highlight the following insights that guide our experiments:

**High structural degrees of freedom in MoE.**  Compared to a dense model (whose shape is almost uniquely determined by $L$, $D_{\mathrm{m}}$, and $\alpha$), an MoE model has many more design choices: the number of MoE layers ($L_{\mathrm{e}}$), expert-related dimensions (*e.g.*, $K$, $E$, $D_{\mathrm{e}}$, $D_{\mathrm{se}}$), and so forth. Even with $L_{\mathrm{d}} = 0$, the final shape depends on $\mu$ and $\beta$ in addition to $\zeta$ and $r_{\mathrm{a}}$. Exhaustively searching all combinations is prohibitively expensive. Therefore, a greedy strategy should be adopted.

**Activation rate $r_{\mathrm{a}}$ is the primary factor.**  At the same total parameter count $N$, the ratio of per-token FLOPs between an MoE model and a dense model is primarily determined by the activation rate $r_{\mathrm{a}}$. More specifically, if $M_{\mathrm{d}}$ is the per-token cost of the dense baseline (with $\zeta, \alpha$ fixed), then the pure MoE model's compute, normalized by $M_{\mathrm{d}}$, roughly behaves as (the union of Equ. 3 and 9):

$$R_{\mathrm{c}} = r_{\mathrm{a}}\left(\frac{4 + 3\alpha + 2\gamma_{\mathrm{d}}}{4 + 3\beta + 2\gamma_{\mathrm{m}}}\right) \tag{10}$$

where $\gamma_{\mathrm{d}}$ and $\gamma_{\mathrm{m}}$ denote ${}^{S}/_{D_{\mathrm{m}}}$ for dense and MoE layers, respectively. As $\gamma$ strongly correlates with $\zeta$, once the shape hyperparameters $(\zeta, \alpha, \beta)$ are chosen, $R_{\mathrm{c}}$ grows monotonically with $r_{\mathrm{a}}$.

**The trade-off among $N$, $C$, and $D$.**  Once $N$ (the total parameters) is chosen and we fix near-optimal shapes for dense and MoE models, the total training compute for the MoE model can be approximated by $C = 3 R_{\mathrm{c}} M_{\mathrm{d}} D$, where $M_{\mathrm{d}}$ is the per-token cost of the dense baseline with the same total parameter count, and $R_{\mathrm{c}}$ is the fraction by which the MoE model reduces compute per token (relative to the dense baseline). If we want to keep the same total compute $C$ for both MoE and dense models, the MoE model will need $R_{\mathrm{c}}$ times more training tokens.

### 3.3 THREE-STEP EXPERIMENTAL METHODOLOGY

Motivated by the aforementioned observations, we propose a three-step experimental methodology which is both comprehensive and fair, so as to achieve the **new perspective** motivated in § 1 that enables a more conclusive comparison of MoE and dense LLMs under equal resource constraints. The methodology is outlined as follows: 1) **Greedy architecture determination.** First, determine the macro-level layer composition (*i.e.*, the MoE-to-dense layer ratio $L_e$ *vs.* $L_d$ and related choices such as shared experts). Second, determine the micro-level MoE design within each MoE layer (*e.g.*, top-$K$ routing and parameter allocation among routed/shared experts), which is largely orthogonal to the global model shape. Finally, select the near-optimal shape hyperparameters (e.g., $\zeta, \alpha$ for dense and $\zeta, \beta$ for MoE) for fair comparisons at a fixed $N$. 2) **Activation rate analysis under fixed $N$ and $C$.** With the optimal MoE architecture chosen in the previous step, and the optimal dense LLM shape proposed by Kaplan et al. (2020), we compare MoE models versus a dense baseline of the same size, ensuring the total training compute $C$ is matched. Since $C$ must be the same, the MoE model typically receives up to $R_c$ times more tokens (initially considering repeated or augmented data). 3) **Data reuse strategy.** To ensure a *truly* fair comparison at the same *unique* data budget $D$, we develop a data reuse strategy that offsets MoE's additional data requirement. This enables evaluation under strictly equal $N$, $C$, and $D$.

### 3.4 COMMON EXPERIMENTAL SETUP

**Optimal hyperparameters.** MoE training is sensitive to the learning rate ($\eta$) and batch size ($B$) (He et al., 2024). Even minor architectural changes, such as variations in $E$, can lead to different optimal hyperparameters. To address this, we train all our models using the optimal $\eta$ and $B$ based on the hyperparameter scaling laws proposed in (Li et al., 2025). Specifically, Li et al. (2025) found that the optimal $\eta$ and $B$ follow power laws and depend only on $N$ and $D$. Since these scaling laws are applicable to both dense and MoE models and are robust across various pretraining data distributions, we apply them to determining $\eta$ and $B$ for each of our experiments.

**Dense baseline tuning.** To ensure a strictly fair comparison, we also tuned the dense baselines by searching for near-optimal structural ratios (e.g., aspect ratio $\zeta$ and FFN expansion $\alpha$, equivalently $L, D_m, D_{ffn}$ with given N), guided by the scaling-law recommendations in Li et al. (2025); the resulting dense configurations used throughout the paper are summarized in Table 13.

**Others.** We use internal, high-quality training and validation datasets composed primarily of diverse web text and specific domains such as mathematics and code. The training and validation sets have different distributions, requiring the evaluated models to demonstrate strong generalization capabilities. Our models incorporate RMSNorm (Zhang & Sennrich, 2019) for pre-normalization, ALiBi (Press et al., 2022) positional encoding for multi-head attention, and the SwiGLU (Shazeer, 2020) activation function for both feed-forward networks (FFNs) and MoE experts. The training procedures used consistently across all experiments are outlined in Table 4 in Appendix E. We employ cross-entropy loss ($\mathcal{L}$) as the training metric and bits-per-character (BPC) as the validation metric.

## 4 OPTIMIZED MOE ARCHITECTURE

Building on the insights discussed above, we systematically examine the following model components in the order outlined below: 1) Distribution of MoE and dense layers. 2) Gate score normalization. 3) Parameter allocation within MoE. 4) Exploration of optimal structural hyperparameters. Each component incorporates previous conclusions into its experimental settings.

**MoE and dense layers arrangement.** This part examines how to arrange the distribution between MoE and dense layers. We consider three layer arrangement schemes: every layer is an MoE layer (`full`), one dense layer followed by MoE layers (`1dense`), and interleaved MoE and dense layers (`interleave`). We additionally include shared experts (SE) for some of our experiments.

Table 5 in Appendix E presents the experimental settings and results. The conclusions are as follows: 1) `1dense`+SE performs the best, possibly because the dense layer contributes to more stable training. 2) The ratio of shared expert size to total expert size $D_{se}/(D_{se} + KD_e)$ has minimal

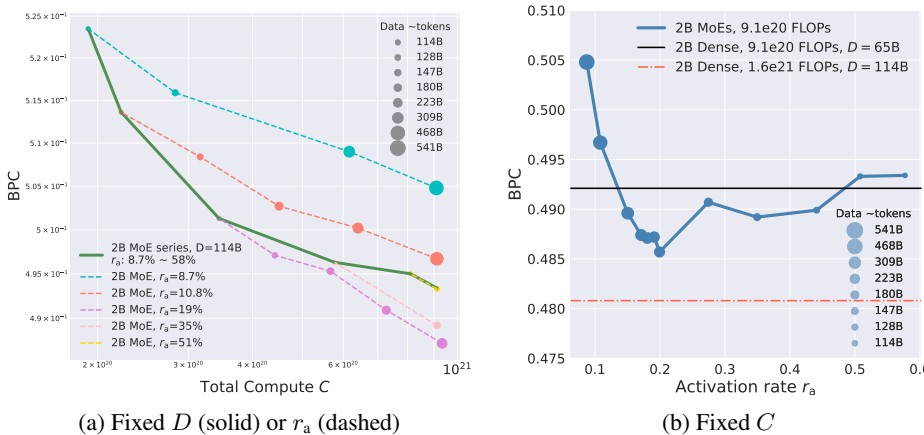

(a) Fixed $D$ (solid) or $r_a$ (dashed)     (b) Fixed $C$

Figure 1: Performance of $N \approx 2$B models trained with varying data sizes $D$ and activation rates $r_a$. **(a)** With a fixed $D$, performance gain exhibits a **non-linear** dependence on training budget $C$. Conversely, with a fixed $r_a$, increasing $D$ leads to a **linear** performance gain. These findings indicate an optimal activation rate, $r_a^{**} = 20\%$, that is consistent across various $D$ values when $N$ is constant. **(b)** With a fixed training compute $C$, the optimal activation rate $r_a^{**} = 20\%$ can be clearly seen.

impact on model performance. Therefore, we continue using the `1dense+SE` configuration and set $D_{se} = KD_e$.

**Gate score normalization.** The results of normalizing gate scores of chosen experts are in Table 6 in Appendix E. Although the addition of normalization does not show an obvious difference in performance loss, it tends to reduce the average balance loss $\bar{\mathcal{L}}_{balance}$. Since normalization requires $K > 1$ to avoid zero gradient, we opt not to normalize given that some of our experiments have $K$=1.

**Top-K setting.** In this part, we discuss the allocation of parameters within MoE layers, focusing on the top-$K$ setting. Expert granularity is adjusted by varying $K$ and $D_e$ while keeping their product constant. We conduct three groups of experiments with various $r_a$ and the results are given in Table 7 in Appendix E. Note that within each experiment group, the product $K \cdot D_e$ is not strictly maintained due to compatibility with other hyperparameters. We observe that both overly large $K$ and the $K$=1 setting are generally suboptimal across the three groups. Therefore, we avoid using large $K$ and avoid setting $K$=1 in our main experiments whenever possible.

**Model shape ratios.** As discussed in § 3.2, the shape hyperparameters include three ratios: $\zeta$, $\alpha$, and $\beta$. We set $\alpha = 2.77$ (Touvron et al., 2023b) and explore the optimal $\zeta$ and $\mu$, from which $\beta$ can be derived. As illustrated in Fig. 4 in Appendix E, although performance fluctuates wildly given a value of $\zeta$ or $\mu$, there is an overall upward trend with increasing $D_m$ for $\zeta$ and a downward trend for $\mu$. Following the observed trend, we set $\zeta \approx 88$ and $\mu \approx 22$ for the subsequent experiments.

## 5 Optimal Activation Rate

In this section, we analyze how the performance of MoE LLMs varies with different activation rates (AR) using model backbones optimized based on the conclusions in § 4, and examine whether MoE models can outperform dense models. Note that a concurrent study (Abnar et al., 2025) suggests that the optimal sparsity of MoE depends on model capacity. However, our findings indicate that, with optimized backbones, the optimal AR remains **consistent** across models of different sizes. We first detail our experimental setup and results, followed by a further discussion on the conclusions.

**Setup.** We built a series of MoE models with non-vocabulary parameters $N \approx 2$B and $N \approx 7$B, but varying activation rates $r_a$ from 8.7% to 58%. Noteworthy, the model backbones were built upon the findings in § 4, as detailed in Table 9, 10, 11 and 12 in Appendix E. Each model was trained on a proportional subset of our dataset, ensuring $D/N \geq 20$ (Hoffmann et al., 2022) for sufficient training.

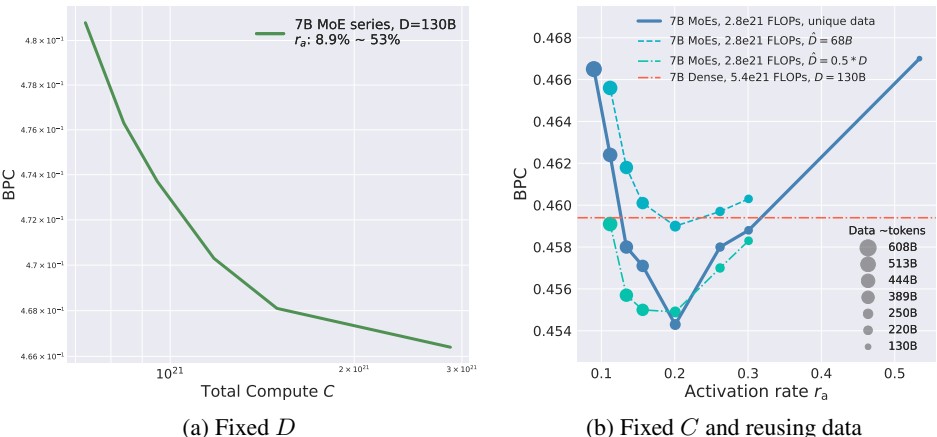

Figure 2: Performance of $N \approx$ 7B models trained with varying data sizes $D$ and activation rate $r_a$. The optimal activation rate, $r_a^{**} = 20\%$, align with the findings for the 2B models (Figure 1). Additionally, compared to training on the unique dataset, the strict data reuse scheme shows only a slight performance reduction, while the loose scheme often yields better performance.

## 5.1 Optimal AR Point

Focusing on the 2B models trained on the same data size $D = 114$B as shown by the green solid line in Figure 1a, we observe that the performance gain **depends non-linearly** on the training budget $C$. Specifically, the gain is more significant within a relatively low range of $r_a$. Starting from points on this curve and fixing the corresponding $r_a$ values, increasing $D$ results in **linearly diminishing** BPC, as indicated by the dashed lines in Figure 1a. These results confirm the **existence of an optimal AR point, $r_a^{**}$, that remains consistent regardless of $D$ when $N$ is unchanged**. When plotting the results from a fixed training compute perspective ($C = C_0 = 9.1e20$) in Figure 1b, we clearly observe that the optimal AR point is approximately $r_a^{**} \approx 20\%$.

## 5.2 Comparison with Dense Models

To compare with dense models, we trained two dense models (Table 13 in Appendix E) with $N \approx 2$B parameters and training budgets $C_1 = C_0 = 9.1e20$ and $C_2 = 1.64e21 \approx 2C_1$. The second model ($C_2$) is included for comparison to account for the typically lower Model FLOPs Utilization (MFU) in MoE training. This reduced MFU arises from load balancing and expert parallelism mechanisms that limit large-block matrix computations. As illustrated in Figure 1b, MoE models outperform their $C_1$ dense counterparts when $r_a$ falls within a specific range (approximately 15% to 48% for 2B models). For instance, the MoE model with the optimal AR point $r_a^{**} = 20\%$ achieves a BPC value that is 0.0064 lower than its $C_1$ dense counterpart and only 0.0049 higher than the $C_2$ dense model. This demonstrates the existence of an *optimal activation rate region* $R_a^*$, where **MoE models with $r_a \in R_a^*$ can outperform their dense counterparts under the same training budget** $C$ and approach the performance of dense models with double the compute. However, the performance gains of MoE models rely on a substantial increase in data, *e.g.*, a $4.6\times$ larger data size at $r_a = r_a^{**} = 20\%$. To mitigate this additional data requirement, we explore a data reuse strategy in § 6.

## 5.3 Consistency of Optimal AR

As illustrated in Fig. 2, an optimal AR point $r_a^{**}$ also exists for 7B models. Surprisingly, $r_a^{**}$ remains consistent for both 2B and 7B models at approximately 20%, suggesting that $r_a^{**}$ **is independent of model size**. This finding contradicts established studies on MoE sparsity (Abnar et al., 2025), which proposes that optimal sparsity (defined as $(E - K)/E$) is directly proportional to model size. Nevertheless, our experiments were conducted with strictly controlled variables using optimized backbones, leading us to believe that our conclusions are both **reliable** and **scalable** (see Appendix B

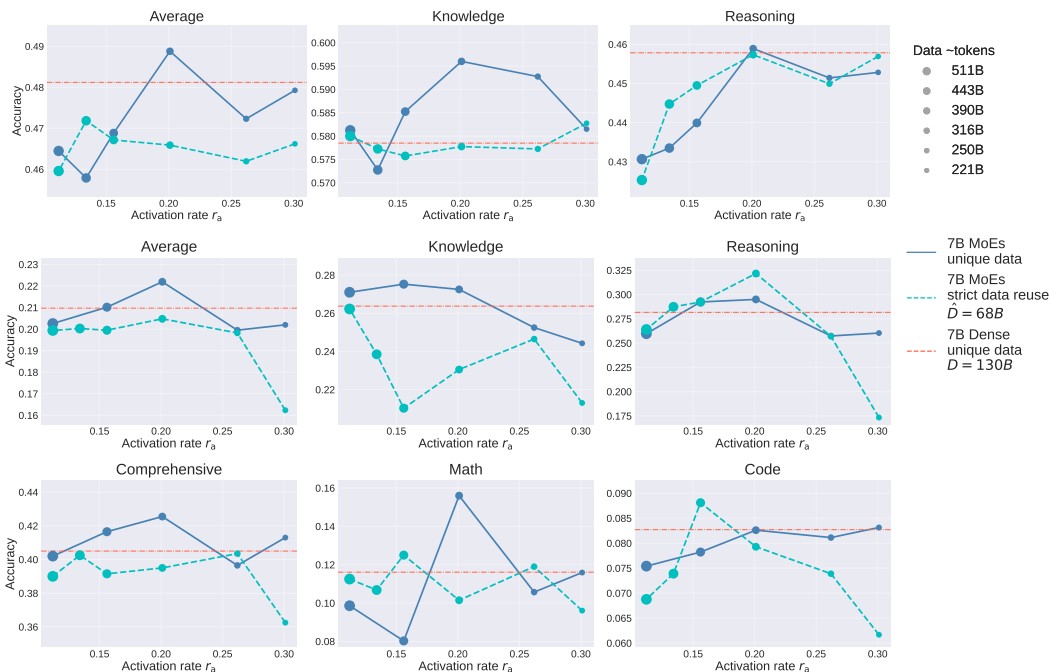

Figure 3: Downstream performance of 7B models: pre-trained (**top**) and SFT-ed (**middle and bottom**) versions. Across all benchmark types, MoE models with $r_a = 20\%$ outperform dense model trained with twice the compute, aligning with upstream observations that the optimal AR is 20%.

for a more detailed discussion). To further validate the possible universality of our findings, we conducted experiments on $N \approx 3B$ models and achieved similar results (see Fig. 5 in Appendix E).

Expert specialization is another significant potential of MoEs in addition to remarkable scalability, where each expert focuses on learning specific features or patterns within the data. However, this attribute has not yet been clearly observed even in state-of-the-art MoE LLMs (Lo et al., 2024; Zhang et al., 2024), and effective approaches to achieve it remain under-explored. Based on our observation that MoEs outperform dense models when $r_a \in R_a^*$, we conjecture a relationship between the optimal AR region and the degree of expert specialization. Specifically: 1) When the activation rate is too low ($r_a < 10\%$), the model lacks sufficient parameters to store knowledge effectively. 2) When the activation rate is relatively high ($r_a > 50\%$), more experts are typically activated, which may lead to weaker specialization. An activation rate within the optimal region $R_a^*$ likely facilitates a higher degree of expert specialization, thereby enhancing the MoE model's performance compared to its dense counterpart. We leave further analysis of this potential relationship for future work.

## 6 DATA REUSE STRATEGY

As discussed in § 5.2, MoEs outperform their dense counterparts but require additional data. To eliminate this increased data demand, we investigate data reusability by training models for multiple epochs using a fixed, smaller dataset size $\hat{D}$. We extract a sub-dataset of size $\hat{D}$ from the original training dataset. At the beginning of each epoch after the first, the data are shuffled.

**Setup.** We explore two distinct schemes, termed the *strict* and the *loose* data reuse schemes.

For the strict scheme, our aim is to ensure that both MoE and dense models are trained under *completely equal conditions with respect to N, D, and C*. Given a fixed $\hat{D}$, the number of training epochs (ranging from 1.7 to 8.3 in our 3B and 7B model experiments) increases as $r_a$ decreases (hence decreasing $M$) to maintain the compute budget $C$. The experimental settings are detailed in Table 14, 15, 16, in Appendix E, which are mostly the same as those in § 5.2, except for the training data used. Specifically, we set $\hat{D} = 65B$ and $114B$ for the 3B models, and $\hat{D} = 68B$ for the 7B models, corresponding to the data used for training the dense models.

Table 2: Accuracy of 7B SFT-ed models across different benchmarks.

| | | Dense baseline | MoE w/ optimal AR | |
|---|---|---|---|---|
| Pretrain info | Activation rate | - | 20.07 | 20.07 |
| | Compute | 5.45e21 | 2.86e21 | 2.86e21 |
| | Data reuse | - | - | strict |
| Knowledge | CMMLU (Li et al., 2024) | 31.23 | 31.62 | **32.11** |
| | MMLU (Hendrycks et al., 2021b) | 31.26 | **32.92** | 24.57 |
| | MMLU-Redux (Gema et al., 2024) | 28.90 | **30.93** | 23.73 |
| | MMLU-Pro (Wang et al., 2024) | **14.12** | 13.59 | 13.59 |
| Reasoning | DROP (Dua et al., 2019) | 32.32 | **35.13** | 30.93 |
| | LiveBench (White et al., 2024) | 16.82 | **18.15** | 16.76 |
| | MUSR (Sprague et al., 2023) | 35.98 | 35.58 | **48.94** |
| Comprehensive | AGIEval (Zhong et al., 2023) | 20.89 | **22.07** | 21.02 |
| | BBH (Suzgun et al., 2023) | 58.02 | **60.01** | 56.07 |
| Math | GAOKAO-Math24 (Zhang et al., 2023) | 9.92 | **15.70** | 9.09 |
| | GSM8K (Cobbe et al., 2021) | 13.34 | **15.54** | 11.22 |
| Code | APPS (Hendrycks et al., 2021a) | 7.35 | 6.80 | **8.18** |
| | DS-1000 (Lai et al., 2023) | 5.70 | **6.90** | 4.60 |
| | HumanEval (Chen et al., 2021) | **22.56** | 21.34 | 21.95 |
| | LeetCode (Coignion et al., 2024) | 1.49 | **1.67** | 1.49 |
| | LiveCodeBench (Jain et al., 2024) | 4.21 | **4.63** | 3.37 |

For the loose scheme, we relax the constraint of identical $D$ by fixing the number of training epochs to 2 for all $r_a$, hence $\hat{D} = 0.5D$, where the exact value of $D$ corresponds to the specific $r_a$. We conduct experiments on 7B models and the experimental settings are in Table 17 in Appendix E.

**Results.** The performance under the strict scheme is illustrated by the blue dashed lines in Fig. 2b and 5 in Appendix E. Reusing data $\hat{D}$ only marginally diminishes performance compared to training on the unique dataset $D$ for a single epoch, and MoE models continue to outperform dense baselines. Moreover, increasing $\hat{D}$ further narrows the performance gap. The similarity in curve shapes indicates that the optimal activation rate $r_a^{**}$ remains unchanged. These findings address the primary question posed at the beginning of this paper: **Mixture-of-Experts can surpass dense LLMs under equal total parameters, compute, and data constraints, provided that the backbones are optimized and $r_a \in R_a^*$.** We further discuss the reuse-vs-unique trade-offs below.

**Discussion.** Prior works have explored the effectiveness of multi-epoch training for dense and MoE models. Muennighoff et al. (2023) developed a scaling law that accounts for the number of repeated tokens and found negligible loss for repeating up to 4 epochs compared to training on unique data, whereas Hernandez et al. (2022) showed degradation for dense models. Xue et al. (2023) noticed no significant gain for MoEs with repeated training when high-quality data is insufficient. We emphasize that our goal here is *not* to claim that multi-epoch training is generally better for MoEs; rather, we examine whether MoEs can still surpass dense models when the *unique-token budget* is fixed. Concretely, under the loose scheme (green dashed line), for each $r_a$ we keep the consumed-token budget $D$ fixed and compare (i) a 2-epoch run on a subset of size $\hat{D} = 0.5D$ (thus processing $D$ tokens with reuse) and (ii) a 1-epoch run that consumes $D$ tokens without reuse (i.e., $D$ unique tokens), where both are sampled from the same data recipe/distribution. We find that the 2-epoch reuse setting can match, and sometimes slightly improve over, the 1-epoch unique-token baseline at several suboptimal $r_a$ points; however, for the 7B models, at the most important optimal point ($r_a \approx 20\%$) it does not exceed the 1-epoch unique-token baseline. In all case for the 7B models, using more than two epochs (multi-epoch) consistently degrades performance. For the 3B models under the strict reuse setting, at a fixed $r_a$, using a larger unique-token budget (e.g., $\hat{D} = 114$B) consistently outperforms a smaller one (e.g., $\hat{D} = 65$B), aligning with the intuition that more unique data is better even under multi-epoch training (see Fig. 5 and Tables 15–16 in Appendix E). Moreover, increasing the unique-token budget from 114B (trained for 2–3 epochs) to 309B (trained for 1 epoch) yields only a marginal improvement under the same token-consumption budget (Fig. 5). Moreover, under a fixed consumed-token budget ($D = 309$B), increasing the unique-token budget from $\hat{D} = 114$B (trained for ∼2–3 epochs with reuse) to 309B (trained for 1 epoch without reuse) yields only a marginal improvement (Fig. 5). Overall, these results suggest that MoE models may

tolerate mild repetition (around two epochs) under a fixed token-consumption budget, but additional repetition becomes harmful.

## 7 ANALYSIS OF DOWNSTREAM PERFORMANCE

To assess whether the optimal ARs generalize to downstream tasks, we conduct SFT on our 7B pre-trained models (trained w/ and w/o strict data reuse) and evaluate both the pre-trained models and SFT-ed models on a total number of 29 benchmarks (see Fig. 3 and Tab. 2), including categories such as `reasoning` and `knowledge`. The comprehensive list of benchmarks can be found in Appendix D. For all SFT trainings, we use a fixed data size $D$, and thus varying $C$ across models with different $r_a$. The 7B dense model trained with *twice* the compute is included for comparison.

**MoE *vs.* Dense at $r_a = r_a^{**}$.** For both PT and SFT models, MoEs outperform their dense equivalents across all benchmark types when $r_a = 20\%$. This result aligns with upstream findings that the optimal activation rate ($r_a^{**}$) is 20%, highlighting the very possible *universality of the optimal AR point across different training phases and data domains*. Furthermore, $r_a^{**}$ remains unchanged during SFT, even with varying $C$, suggesting that the SFT data size may have an upper limit for performance improvement, provided the PT model is adequately trained. Additionally, the `average` performance at $r_a \neq r_a^{**}$ is consistently inferior to that of dense models, underscoring the critical role of the optimal AR point.

**MoE *vs.* Dense at $r_a < r_a^{**}$.** Dense models outperform MoEs across all domains for PT models. After SFT, MoEs overtake dense models on `comprehensive` and `knowledge` tasks. However, a notable performance gap remains in `math`, highlighting the PT stage's importance for math ability.

**MoE *vs.* Dense at $r_a > r_a^{**}$.** Compared to the dense models, MoEs perform better on `knowledge` but worse on `reasoning` for PT models, and usually slightly underperform after SFT.

**Sparser *vs.* Denser.** For PT models, denser MoEs ($r_a > r_a^{**}$) outperform or match the performance of sparser MoEs ($r_a < r_a^{**}$) across all domains, consistent with Figure 2b, regardless of data reuse. When training on unique data, sparser MoEs perform better on `knowledge`. Notably, denser MoE performance significantly degrades with data reuse, especially for SFT.

**Impact of data reuse.** For both PT and SFT models, data reuse has little impact on `reasoning` but causes significant degradation in `knowledge` performance. Surprisingly, at $r_a = r_a^{**}$, the SFT-ed MoEs trained with data reuse outperform both MoEs and dense models trained on unique data. This implies that *a model can master reasoning skills (rather than merely memorizing information (Hu et al., 2024)) with a relatively small dataset (Muennighoff et al., 2025; Wang et al., 2025) and further enhance its capabilities through multiple training epochs*.

## 8 CONCLUSION AND FUTURE WORKS

In this paper, we propose a three-step experimental methodology to investigate whether MoEs can surpass their dense counterparts under the same constraints on total parameters, compute, and data. By optimizing the architecture, identifying the optimal activation rate region, and reusing data, we arrive at a positive answer to this question. Future work will explore how optimal activation rates enhance model capabilities and whether similar conclusions hold for other training methods like upcycling (Komatsuzaki et al., 2023) and MoEfication (Zhang et al., 2022). We hope this work offers valuable insights for the architectural design of next-generation models.

### LIMITATIONS

The limitations of this work include: 1) Hindering by the high computational cost, we did not train models larger than 7B. 2) As described in § 4, we focus mainly on the impact of several main components of MoEs, but fix the rest to narrow the scale of experiments. 3) Exploration of other elements can provide further comprehensive guidance for the architectural design of MoEs.

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

# APPENDIX

## A  BACKGROUND: MIXTURE-OF-EXPERTS

The MoE architecture primarily consists of a gate and several experts. Typically, the gate $g(\cdot)$ is composed of a linear layer $W_g$ followed by a Softmax and a Top-K operation, and the experts $\{E_i\}_{i=1}^E$ follow the standard FFN structure. The computation of an MoE block can then be formulated as

follows:

$$s_i(x) = \text{Softmax}_i(W_g x), \tag{11}$$

$$\mathcal{T}(x) = \text{TopK}(s(x); K), \tag{12}$$

$$g_i(x) = \begin{cases} s_i(x) & \text{if } i \in \mathcal{T}(x), \\ 0 & \text{otherwise,} \end{cases} \tag{13}$$

$$y = \sum_{i=1}^{E} g_i(x) \cdot E_i(x), \tag{14}$$

where $s_i(x)$ denotes the gate score for the $i$-th expert. Unless otherwise specified, we use the above *non-normalized* Top-$K$ gating in this paper (*i.e.*, we do not renormalize the $K$ selected scores to sum to 1). For completeness, the commonly used *Top-$K$ normalization* is

$$\tilde{g}_i(x) = \begin{cases} \dfrac{g_i(x)}{\sum_{j \in \mathcal{T}(x)} g_j(x)} & \text{if } i \in \mathcal{T}(x), \\ 0 & \text{otherwise.} \end{cases} \tag{15}$$

We also adopt the standard auxiliary load-balancing loss (used throughout our experiments) to encourage uniform expert utilization. Given a minibatch $\mathcal{B}$, define

$$f_i = \frac{1}{|\mathcal{B}|} \sum_{x \in \mathcal{B}} \mathbb{I}[i \in \mathcal{T}(x)], \tag{16}$$

$$p_i = \frac{1}{|\mathcal{B}|} \sum_{x \in \mathcal{B}} s_i(x), \tag{17}$$

and compute

$$\mathcal{L}_{\text{balance}} = E \sum_{i=1}^{E} f_i\, p_i, \qquad \mathcal{L}_{\text{total}} = \mathcal{L}_{\text{CE}} + \lambda \mathcal{L}_{\text{balance}}. \tag{18}$$

## B  EXTENDED ANALYSIS ON RELATED WORK

We notice a concurrent work (Abnar et al., 2025) studied scaling law for optimal MoE sparsity. We highlight the differences between our work and theirs as follows:

- **Formulation**: We define "sparsity" as the activation rate $r_a = {N_a}/{N}$, which is a more general definition than that proposed by Abnar et al. (2025), namely the ratio of inactive experts to the total number of experts, ${(E - K)}/{E}$.

- **Methodology**: Given that the activation rate $r_a$ does not depend on the underlying model architecture, we can thus easily take into consideration other components such as shared expert and build all our models upon the optimized architecture proposed in § 4. This ensures the observed performance differences solely attribute to the varying activation rates.

- **Sufficient training**: Our main comparisons operate in a sufficiently trained regime (often beyond the compute-optimal point). For example, the 2B MoE models in Table 9 have ${D}/{N}$ ranging from 53 to 252, and the 7B MoE models in Table 11 have ${D}/{N}$ ranging from 20 to 93, while our dense baselines satisfy ${D}/{N} \geq 20$ (Table 13), aligning with the common compute-optimal guideline (Hoffmann et al., 2022). This perspective is practically important since industrial model design often cares about the best achievable performance at a fixed-$N$ budget (e.g., training a 7B model on trillions of tokens), and it also enables more reliable downstream SFT comparisons that would be less meaningful with undertrained checkpoints.

- **Perspective under limited resources**: Scaling-law studies often adopt broad sweeps that trade depth for breadth under limited compute, which can lead to undertrained large-model settings (e.g., Abnar et al. (2025) uses $C = 1\mathrm{e}20$ per run for a sweep up to 30B, and Ludziejewski et al. (2025) uses at most 80B tokens). In contrast, our 7B study allocates substantially more compute per setting (e.g., multiple runs with $C \approx 2.86\mathrm{e}21$ or $C \approx 5.45\mathrm{e}21$; Table 12 and Table 13), prioritizing a high-${D}/{N}$ regime to more deeply investigate the fixed-$N$ & fixed-$C$ question motivated by deployment memory constraints.

Table 3: Pretraining data recipe compared with the LLaMA-1 recipe.

| DataSet Class | Our Recipe | Our Data Set Detail | LLaMA-1 Recipe | LLaMA-1 Data Set Detail | Recipe Diff |
|---|---|---|---|---|---|
| WebData-en | 79.53% | CC (English) | 82% | 67% CC + 15% C4 (English) | -2.47% |
| Code | 4.62% | The Stack | 4.50% | Github-Big Query | +0.12% |
| Wikipedia | 5.06% | en: 1.69%, cn: 0.13%, others: 3.24% | 4.50% | multi-lingual | +0.56% |
| Book | 5.18% | open source English books | 4.50% | book3, Gutenberg | +0.68% |
| arXiv | 3.38% | as class name | 1.06% | as class name | +2.32% |
| StackExchange | 2.21% | as class name | 2.00% | as class name | +0.21% |

- **Conclusion**: We discover an optimal activation rate that appears to be *independent* of model sizes, whereas Abnar et al. (2025) find that the optimal sparsity increases with model size.

Our conclusion regarding a consistent optimal activation rate contradicts the findings of Abnar et al. (2025). While we believe our findings are reliable, given that our experiments are conducted with strictly controlled variables using *optimized backbones* and *sufficient training data*, we acknowledge the possibility that the optimal $r_a$ might slightly shift for model sizes significantly beyond our studied range (*i.e.*, $N >> 7B$). Nevertheless, we contend that the optimal $r_a$ can be considered consistent within a certain range of model sizes, in contrast to the significant changes reported by Abnar et al. (2025).

## C  PRETRAINING DATA RECIPE

For reproducibility, we provide the mixture ratios of our pretraining corpus and a comparison with the LLaMA-1 recipe (Touvron et al., 2023a). Our recipe is intentionally close to a LLaMA-1–style mixture, and the corresponding data sources have public counterparts.

## D  COMPREHENSIVE LIST OF BENCHMARKS

To assess whether the optimal ARs generalize to downstream tasks, we conduct SFT on our 7B pre-trained models (trained w/ and w/o strict data reuse) and evaluate both the pre-trained models and SFT-ed models on a total number of 29 benchmarks (Figure 3). The comprehensive list of benchmarks is provided here.

For pre-trained models, we evaluate on:

- Knowledge: BBH (Suzgun et al., 2023), PIQA (Bisk et al., 2019), SCIQ (Welbl et al., 2017), SIQA (Sap et al., 2019)
- Reasoning: ARC (Clark et al., 2018), BoolQ (Clark et al., 2019), CLUE (Xu et al., 2020), DROP (Dua et al., 2019), HellaSwag (Zellers et al., 2019), NaturalQA (Kwiatkowski et al., 2019), RACE (Lai et al., 2017), WinoGrande (Sakaguchi et al., 2020), XTREME (Hu et al., 2020)

For SFT-ed models, we evaluate on:

- Comprehensive: AGIEVAL (Zhong et al., 2023), BBH
- Knowledge: CMMLU (Li et al., 2024), MMLU (Hendrycks et al., 2021b), MMLU-Redux (Gema et al., 2024), MMLU-Pro (Wang et al., 2024)
- Reasoning: DROP, LiveBench (White et al., 2024), MuSR (Sprague et al., 2023)
- Math: GAOKAO-Math24 (Zhang et al., 2023), GSM8K (Cobbe et al., 2021)
- Code: APPS (Hendrycks et al., 2021a), DS-1000 (Lai et al., 2023), HumanEval (Chen et al., 2021), LeetCode (Coignion et al., 2024), LiveCodeBench (Jain et al., 2024)

## E  MORE EXPERIMENTAL RESULTS

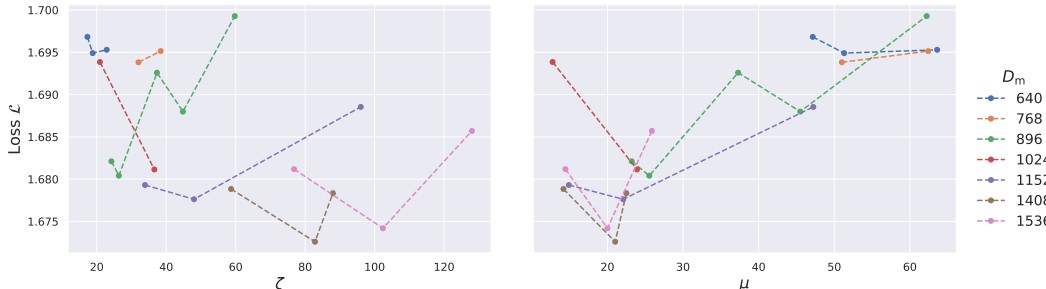

Figure 4: Results for model shape ratios $\zeta$ and $\mu$. An overall upward trend is observed in $\zeta$ as $D_{\mathrm{m}}$ increases, while $\mu$ exhibits a downward trend with increasing $D_{\mathrm{m}}$.

Table 4: Common training recipe.

| Hyperparameter | Setting |
|---|---|
| Vocab | 65536 |
| Optimizer | Adam |
| Weight decay | 0.1 |
| Gradient clipping norm | 1.0 |
| LR Scheduler | Cosine |
| Warmup iters | $\mathrm{clip}(0.01 \cdot \mathrm{Iters}, 200, 2000)$ |
| Min LR | 1e-5 |

Table 5: Experimental settings and results of MoE layer arrangement and shared expert. Hyperparameters shared by all experiments: $D_{\mathrm{m}} = 1408, D_{\mathrm{ffn}} = 3904, \mathrm{Norm} = \mathrm{True}$.

| $N$ | $N_{\mathrm{a}}$ | $M$ | $H$ | $D_{\mathrm{h}}$ | $L$ | $E$ | $K$ | $D_{\mathrm{e}}$ | $D_{\mathrm{se}}$ | Scheme | $\mathcal{L}$ | Conclusion |
|---|---|---|---|---|---|---|---|---|---|---|---|---|
| 2.02B | 346M | 8.77e8 | 22 | 64 | 16 | 35 | 2 | 800 | 1600 | `full`+SE | 1.6813 | |
| 2.02B | 346M | 8.77e8 | 22 | 64 | 16 | 68 | 2 | 800 | 1600 | `interleave`+SE | 1.6766 | `interleave` performs better than `full` |
| 2.02B | 346M | 8.77e8 | 22 | 64 | 16 | 70 | 4 | 800 | 0 | `interleave` | 1.6697 | |
| 2.15B | 366M | 6.63e9 | 11 | 128 | 16 | 85 | 5 | 352 | 1760 | `1dense`+SE | 1.8700 | |
| 2.15B | 366M | 6.63e9 | 22 | 64 | 16 | 85 | 5 | 352 | 1760 | `1dense`+SE | 1.8557 | `1dense`+SE performs the best |
| 2.15B | 367M | 6.63e9 | 11 | 128 | 16 | 70 | 4 | 800 | 0 | `interleave` | 1.8737 | |
| 2.15B | 367M | 6.63e9 | 22 | 64 | 16 | 70 | 4 | 800 | 0 | `interleave` | 1.8620 | |
| 2.15B | 368M | 9.31e8 | 22 | 64 | 17 | 37 | 4 | 800 | 0 | `1dense` | 1.6752 | |
| 2.15B | 368M | 9.31e8 | 22 | 64 | 17 | 36 | 3 | 800 | 800 | `1dense`+SE | 1.6712 | $\frac{D_{\mathrm{se}}}{(D_{\mathrm{se}} + K D_{\mathrm{e}})}$ impacts little |
| 2.15B | 368M | 9.31e8 | 22 | 64 | 17 | 35 | 2 | 800 | 1600 | `1dense`+SE | 1.6726 | |

Table 6: Experimental settings and results of gate score normalization. Hyperparameters shared by all experiments: $\mathrm{Scheme} = \mathtt{1dense}, L = 17, D_{\mathrm{m}} = 1408, D_{\mathrm{ffn}} = 3904, H = 22, D_{\mathrm{h}} = 64$.

| $N$ | $N_{\mathrm{a}}$ | $r_{\mathrm{a}}$ (%) | $M$ | $E$ | $K$ | $D_{\mathrm{e}}$ | $D_{\mathrm{se}}$ | Norm | $\mathcal{L}$ | $\bar{\mathcal{L}}_{\mathrm{balance}}$ |
|---|---|---|---|---|---|---|---|---|---|---|
| 2.15B | 368M | 17.08 | 9.31e8 | 35 | 2 | 800 | 1600 | Y | 1.6726 | **1.355** |
| 2.15B | 368M | 17.08 | 9.31e8 | 35 | 2 | 800 | 1600 | N | 1.6712 | 1.452 |
| 2.15B | 368M | 17.08 | 9.31e8 | 37 | 4 | 800 | 0 | Y | 1.6752 | **1.409** |
| 2.15B | 368M | 17.08 | 9.31e8 | 37 | 4 | 800 | 0 | N | 1.6750 | 1.440 |

Table 7: Experimental settings and results of top-K setting. Hyperparameters shared by all experiments: $\mathrm{Scheme} = \mathtt{1dense}, L = 16, D_{\mathrm{m}} = 1408, D_{\mathrm{ffn}} = 3904, H = 11, D_{\mathrm{h}} = 128, \mathrm{Norm} = \mathrm{False}$.

| $N$ | $N_{\mathrm{a}}$ | $r_{\mathrm{a}}$ (%) | $M$ | $E$ | $K$ | $D_{\mathrm{e}}$ | $D_{\mathrm{se}}$ | $\mathcal{L}$ |
|---|---|---|---|---|---|---|---|---|
| 2.15B | 591M | 27.47 | 8.00e9 | 8 | 1 | 3528 | 3528 | 2.0470 |
| 2.15B | 591M | 27.40 | 8.00e9 | 88 | 11 | 320 | 3520 | **2.0338** |
| 2.15B | 949M | 44.00 | 1.01e10 | 8 | 2 | 3176 | 6352 | **1.9996** |
| 2.15B | 948M | 44.05 | 1.01e10 | 88 | 22 | 288 | 6336 | 2.0266 |
| 2.15B | 1.24B | 57.57 | 1.19e10 | 8 | 3 | 2888 | 8664 | **2.0156** |
| 2.11B | 1.22B | 57.68 | 1.18e10 | 88 | 33 | 256 | 8448 | 2.0235 |

Table 8: Experimental settings and results of model shape ratios. Hyperparameters shared by all experiments: Scheme = `1dense`, $S = 16384, D_h = 128$.

| $N$ | $N_a$ | $D_m$ | $D_{ffn}$ | $L$ | $H$ | $E$ | $K$ | $D_e$ | $D_{se}$ | $\mu$ | $\zeta$ | $\mathcal{L}$ |
|---|---|---|---|---|---|---|---|---|---|---|---|---|
| 2.15e9 | 3.67e8 | 640 | 1774 | 34 | 5 | 50 | 4 | 608 | 2432 | 51.30 | 20.39 | 1.694 |
| 2.15e9 | 3.69e8 | 640 | 1774 | 37 | 5 | 38 | 3 | 736 | 2208 | 47.15 | 18.78 | 1.696 |
| 2.14e9 | 3.69e8 | 640 | 1774 | 49 | 5 | 41 | 3 | 512 | 1536 | 35.20 | 14.33 | 1.695 |
| 2.14e9 | 3.67e8 | 768 | 2129 | 20 | 6 | 99 | 8 | 448 | 3584 | 62.42 | 41.42 | 1.693 |
| 2.15e9 | 3.69e8 | 896 | 2484 | 15 | 7 | 124 | 10 | 416 | 4160 | 62.21 | 65.00 | 1.699 |
| 2.15e9 | 3.68e8 | 896 | 2484 | 20 | 7 | 91 | 7 | 416 | 2912 | 45.50 | 48.16 | 1.687 |
| 2.13e9 | 3.67e8 | 896 | 2484 | 24 | 7 | 54 | 4 | 576 | 2304 | 37.29 | 39.96 | 1.692 |
| 2.15e9 | 3.69e8 | 896 | 2484 | 34 | 7 | 61 | 4 | 352 | 1408 | 25.54 | 28.15 | 1.680 |
| 2.16e9 | 3.68e8 | 896 | 2484 | 37 | 7 | 47 | 3 | 416 | 1248 | 23.21 | 25.89 | 1.682 |
| 2.14e9 | 3.70e8 | 1024 | 2839 | 28 | 8 | 80 | 5 | 288 | 1440 | 23.91 | 38.93 | 1.681 |
| 2.16e9 | 3.69e8 | 1024 | 2839 | 49 | 8 | 49 | 2 | 256 | 512 | 12.75 | 22.33 | 1.693 |
| 2.15e9 | 3.67e8 | 1152 | 3194 | 12 | 9 | 79 | 6 | 640 | 3840 | 47.22 | 105.73 | 1.688 |
| 2.14e9 | 3.68e8 | 1152 | 3194 | 34 | 9 | 64 | 3 | 256 | 768 | 14.89 | 35.91 | 1.679 |
| 2.15e9 | 3.69e8 | 1280 | 3549 | 28 | 10 | 113 | 5 | 160 | 800 | 14.75 | 48.41 | 1.675 |
| 2.15e9 | 3.70e8 | 1408 | 3904 | 24 | 11 | 46 | 2 | 416 | 832 | 14.18 | 62.22 | 1.678 |
| 2.13e9 | 3.68e8 | 1536 | 4258 | 12 | 12 | 65 | 4 | 576 | 2304 | 25.88 | 140.64 | 1.685 |
| 2.16e9 | 3.68e8 | 1536 | 4258 | 15 | 12 | 91 | 5 | 320 | 1600 | 20.00 | 110.71 | 1.674 |
| 2.15e9 | 3.66e8 | 1536 | 4258 | 20 | 12 | 95 | 4 | 224 | 896 | 14.44 | 81.84 | 1.681 |
| 2.16e9 | 3.68e8 | 1792 | 4968 | 15 | 14 | 128 | 5 | 192 | 960 | 14.25 | 129.00 | 1.693 |
| 2.14e9 | 3.67e8 | 1920 | 5323 | 12 | 15 | 71 | 3 | 416 | 1248 | 16.03 | 175.55 | 1.699 |

Table 9: Experimental settings and results of optimal ARs for MoE models with $N = 2.15B$ and fixed $r_a$. Hyperparameters shared by all experiments: $L = 16, S = 2048, D_m = 1408, D_{ffn} = 3904, H = 11, D_h = 128, \zeta = 88$.

| $N_a$ | $r_a$ (%) | $M$ | $D$ | $C$ | $D/N$ | $E$ | $K$ | $D_e$ | $D_{se}$ | $\eta$ | $B$ | # Iters | BPC |
|---|---|---|---|---|---|---|---|---|---|---|---|---|---|
| 1.88e8 | 8.74 | 1.68e9 | 1.14e11 | 1.92e20 | 53 | 89 | 1 | 352 | 352 | 2.01e-3 | 672 | 82833 | 0.5235 |
| 1.88e8 | 8.74 | 1.68e9 | 1.68e11 | 2.83e20 | 78 | 89 | 1 | 352 | 352 | 2.26e-3 | 832 | 98771 | 0.5159 |
| 1.88e8 | 8.74 | 1.68e9 | 3.67e11 | 6.16e20 | 170 | 89 | 1 | 352 | 352 | 2.87e-3 | 1344 | 133187 | 0.5090 |
| 1.88e8 | 8.74 | 1.68e9 | 5.41e11 | 9.10e20 | 252 | 89 | 1 | 352 | 352 | 3.24e-3 | 1728 | 152927 | 0.5048 |
| 2.33e8 | 10.81 | 1.95e9 | 1.14e11 | 2.22e20 | 53 | 88 | 2 | 352 | 704 | 2.01e-3 | 672 | 82833 | 0.5136 |
| 2.33e8 | 10.81 | 1.95e9 | 1.62e11 | 3.16e20 | 75 | 88 | 2 | 352 | 704 | 2.24e-3 | 896 | 88446 | 0.5084 |
| 2.33e8 | 10.81 | 1.95e9 | 2.31e11 | 4.50e20 | 107 | 88 | 2 | 352 | 704 | 2.49e-3 | 1024 | 110149 | 0.5027 |
| 2.33e8 | 10.81 | 1.95e9 | 3.29e11 | 6.41e20 | 153 | 88 | 2 | 352 | 704 | 2.78e-3 | 1280 | 125427 | 0.5002 |
| 2.33e8 | 10.81 | 1.95e9 | 4.68e11 | 9.12e20 | 218 | 88 | 2 | 352 | 704 | 3.10e-3 | 1600 | 142822 | 0.4967 |
| 4.11e8 | 19.11 | 3.02e9 | 1.14e11 | 3.44e20 | 53 | 84 | 6 | 352 | 2112 | 2.01e-3 | 672 | 82833 | 0.5013 |
| 4.11e8 | 19.11 | 3.02e9 | 1.46e11 | 4.42e20 | 68 | 84 | 6 | 352 | 2112 | 2.17e-3 | 768 | 93015 | 0.4971 |
| 4.11e8 | 19.11 | 3.02e9 | 1.88e11 | 5.67e20 | 87 | 84 | 6 | 352 | 2112 | 2.34e-3 | 960 | 95469 | 0.4953 |
| 4.11e8 | 19.11 | 3.02e9 | 2.41e11 | 7.27e20 | 112 | 84 | 6 | 352 | 2112 | 2.52e-3 | 1024 | 114870 | 0.4909 |
| 4.11e8 | 19.11 | 3.02e9 | 3.09e11 | 9.34e20 | 144 | 84 | 6 | 352 | 2112 | 2.73e-3 | 1280 | 117950 | 0.4872 |
| 7.52e8 | 34.95 | 5.06e9 | 1.14e11 | 5.77e20 | 53 | 84 | 15 | 320 | 4800 | 2.01e-3 | 672 | 82833 | 0.4963 |
| 7.52e8 | 34.95 | 5.06e9 | 1.80e11 | 9.13e20 | 84 | 84 | 15 | 320 | 4800 | 2.31e-3 | 896 | 98256 | 0.4892 |
| 1.09e9 | 50.79 | 7.11e9 | 1.14e11 | 8.10e20 | 53 | 84 | 26 | 288 | 7488 | 2.01e-3 | 672 | 82833 | 0.4950 |
| 1.09e9 | 50.79 | 7.11e9 | 1.28e11 | 9.13e20 | 60 | 84 | 26 | 288 | 7488 | 2.08e-3 | 704 | 89125 | 0.4933 |

Table 10: Experimental settings and results of optimal ARs for MoE models $N = 2.15B$ with fixed $C$. Hyperparameters shared by all experiments: $L = 16, S = 2048, D_m = 1408, D_{ffn} = 3904, H = 11, D_h = 128, \zeta = 88$. The green row corresponds to the MoE model with the lowest BPC on the validation set.

| $N_a$ | $r_a$(%) | $M$ | $D$ | $C$ | $D/N$ | $\mu$ | $E$ | $K$ | $D_e$ | $D_{se}$ | $\eta$ | $B$ | # Iters | BPC |
|---|---|---|---|---|---|---|---|---|---|---|---|---|---|---|
| 1.88e8 | 8.74 | 1.70e9 | 5.41e11 | 9.18e20 | 252 | 22.50 | 89 | 1 | 352 | 352 | 3.24e-3 | 1728 | 152927 | 0.5048 |
| 2.33e8 | 10.82 | 1.96e9 | 4.68e11 | 9.19e20 | 218 | 22.50 | 88 | 2 | 352 | 704 | 3.10e-3 | 1600 | 142822 | 0.4967 |
| 3.24e8 | 15.04 | 2.50e9 | 3.75e11 | 9.38e20 | 174 | 22.50 | 86 | 4 | 352 | 1408 | 2.89e-3 | 1344 | 136378 | 0.4896 |
| 3.68e8 | 17.11 | 2.77e9 | 3.39e11 | 9.38e20 | 158 | 22.50 | 85 | 5 | 352 | 1760 | 2.80e-3 | 1296 | 127756 | 0.4874 |
| 3.89e8 | 18.06 | 2.89e9 | 3.25e11 | 9.38e20 | 151 | 22.50 | 93 | 6 | 320 | 1920 | 2.76e-3 | 1280 | 123862 | 0.4871 |
| 4.11e8 | 19.12 | 3.03e9 | 3.09e11 | 9.38e20 | 144 | 22.50 | 84 | 6 | 352 | 2112 | 2.73e-3 | 1280 | 117950 | 0.4872 |
| 4.29e8 | 19.94 | 3.13e9 | 2.99e11 | 9.38e20 | 139 | 22.50 | 92 | 7 | 320 | 2240 | 2.70e-3 | 1248 | 117177 | **0.4857** |
| 5.90e8 | 27.46 | 4.10e9 | 2.23e11 | 9.15e20 | 104 | 22.55 | 8 | 1 | 3528 | 3528 | 2.46e-3 | 1024 | 106335 | 0.4907 |
| 7.52e8 | 34.96 | 5.08e9 | 1.80e11 | 9.16e20 | 84 | 22.50 | 84 | 15 | 320 | 4800 | 2.31e-3 | 896 | 98256 | 0.4892 |
| 9.48e8 | 44.11 | 6.25e9 | 1.47e11 | 9.16e20 | 68 | 22.56 | 8 | 2 | 3176 | 6352 | 2.16e-3 | 768 | 93460 | 0.4899 |
| 1.09e9 | 50.80 | 7.12e9 | 1.29e11 | 9.15e20 | 60 | 22.50 | 84 | 26 | 288 | 7488 | 2.08e-3 | 704 | 89125 | 0.4933 |
| 1.24e9 | 57.73 | 8.01e9 | 1.14e11 | 9.13e20 | 53 | 22.56 | 8 | 3 | 2888 | 8664 | 2.006e-3 | 672 | 82833 | 0.4934 |

Table 11: Experimental settings and results of optimal ARs for MoE models with $N = 6.52$B with fixed $D$. Hyperparameters shared by all experiments: $L = 24, S = 2048, D_\mathrm{m} = 2048, D_\mathrm{ffn} = 5464, H = 16, D_\mathrm{h} = 128, \zeta = 85.3$.

| $N_\mathrm{a}$ | $r_\mathrm{a}$ (%) | $M$ | $D$ | $C$ | $D/N$ | $E$ | $K$ | $D_\mathrm{e}$ | $D_\mathrm{se}$ | $\eta$ | $B$ | # Iters | BPC |
|---|---|---|---|---|---|---|---|---|---|---|---|---|---|
| 7.26e8 | 11.15 | 5.59e9 | 1.30e11 | 7.25e20 | 19.90 | 82 | 2 | 512 | 1024 | 4.74e-4 | 640 | 98816 | 0.4808 |
| 8.70e8 | 13.36 | 6.46e9 | 1.30e11 | 8.37e20 | 19.90 | 81 | 3 | 512 | 1536 | 4.74e-4 | 640 | 98816 | 0.4763 |
| 1.02e9 | 15.67 | 7.33e9 | 1.30e11 | 9.49e20 | 19.90 | 80 | 4 | 512 | 2048 | 4.74e-4 | 640 | 98816 | 0.4737 |
| 1.31e9 | 20.03 | 9.07e9 | 1.30e11 | 1.17e21 | 19.88 | 78 | 6 | 512 | 3072 | 4.74e-4 | 640 | 98816 | 0.4703 |
| 1.70e9 | 26.11 | 1.15e10 | 1.30e11 | 1.48e21 | 19.90 | 86 | 10 | 448 | 4480 | 4.74e-4 | 640 | 98816 | 0.4681 |
| 3.47e9 | 53.30 | 2.21e10 | 1.30e11 | 2.86e21 | 19.90 | 84 | 28 | 384 | 10752 | 4.74e-4 | 640 | 98816 | 0.4664 |

Table 12: Experimental settings and results of optimal ARs for MoE models with $N = 6.52$B with fixed $C$. Hyperparameters shared by all experiments: $L = 24, S = 2048, D_\mathrm{m} = 2048, D_\mathrm{ffn} = 5464, H = 16, D_\mathrm{h} = 128, \zeta = 85.3$. The green row corresponds to the MoE model with the lowest BPC on the validation set.

| $N_\mathrm{a}$ | $r_\mathrm{a}$(%) | $M$ | $D$ | $C$ | $D/N$ | $\mu$ | $E$ | $K$ | $D_\mathrm{e}$ | $D_\mathrm{se}$ | $\eta$ | $B$ | # Iters | BPC |
|---|---|---|---|---|---|---|---|---|---|---|---|---|---|---|
| 5.85e8 | 8.97 | 4.73e9 | 6.05e11 | 2.86e21 | 92.88 | 21.00 | 83 | 1 | 512 | 512 | 7.62e-4 | 1512 | 195502 | 0.4665 |
| 7.30e8 | 11.19 | 5.59e9 | 5.11e11 | 2.86e21 | 78.47 | 21.00 | 82 | 2 | 512 | 1024 | 7.23e-4 | 1360 | 183630 | 0.4624 |
| 8.74e8 | 13.41 | 6.46e9 | 4.43e11 | 2.86e21 | 67.93 | 21.00 | 81 | 3 | 512 | 1536 | 6.92e-4 | 1232 | 175482 | 0.4580 |
| 1.02e9 | 15.63 | 7.33e9 | 3.90e11 | 2.86e21 | 59.89 | 21.00 | 80 | 4 | 512 | 2048 | 6.64e-4 | 1152 | 165447 | 0.4571 |
| 1.31e9 | 20.07 | 9.07e9 | 3.16e11 | 2.86e21 | 48.50 | 21.00 | 78 | 6 | 512 | 3072 | 6.23e-4 | 1040 | 148410 | **0.4543** |
| 1.71e9 | 26.18 | 1.15e10 | 2.50e11 | 2.86e21 | 38.32 | 21.00 | 86 | 10 | 448 | 4480 | 5.80e-4 | 960 | 127035 | 0.4580 |
| 1.96e9 | 30.07 | 1.30e10 | 2.21e11 | 2.86e21 | 33.83 | 21.00 | 84 | 12 | 448 | 5376 | 5.57e-4 | 800 | 134597 | 0.4588 |
| 3.48e9 | 53.38 | 2.21e10 | 1.30e11 | 2.86e21 | 19.87 | 21.00 | 84 | 28 | 384 | 10752 | 4.74e-4 | 640 | 98816 | 0.4670 |

Table 13: Experimental settings and results of optimal ARs for 2B, 3B, and 7B dense baselines.

| $N$ | $M$ | $D$ | $C$ | $D/N$ | $L$ | $H$ | $D_\mathrm{m}$ | $D_\mathrm{ffn}$ | $\eta$ | $B$ | # Iters | BPC |
|---|---|---|---|---|---|---|---|---|---|---|---|---|
| 2.15e9 | 1.44e10 | 6.50e10 | 9.36e20 | 30.23 | 28 | 17 | 2176 | 8848 | 8.44e-4 | 320 | 99182 | 0.4921 |
| 2.15e9 | 1.44e10 | 1.14e11 | 1.64e21 | 53.02 | 28 | 17 | 2176 | 8848 | 1.00e-3 | 448 | 124032 | 0.4808 |
| 3.29e9 | 2.24e10 | 6.26e10 | 1.40e21 | 19.03 | 44 | 19 | 2432 | 7008 | 1.23e-3 | 448 | 68253 | 0.4833 |
| 3.29e9 | 2.24e10 | 1.25e11 | 2.80e21 | 38.06 | 44 | 19 | 2432 | 7008 | 1.52e-3 | 640 | 95554 | 0.4684 |
| 6.48e9 | 4.21e10 | 6.80e10 | 2.86e21 | 10.49 | 32 | 32 | 4096 | 11008 | 3.89e-4 | 432 | 76813 | 0.4736 |
| 6.48e9 | 4.21e10 | 1.30e11 | 5.45e21 | 20.00 | 32 | 32 | 4096 | 11008 | 4.76e-4 | 640 | 98816 | 0.4594 |

Table 14: Experimental settings and results of data reusing ($\hat{D} = 68$B) for MoE models with $N = 6.52$B with fixed $C$. Hyperparameters shared by all experiments: $L = 24, S = 2048, D_\mathrm{m} = 2048, D_\mathrm{ffn} = 5464, H = 16, D_\mathrm{h} = 128, \zeta = 85.3$. The green row corresponds to the MoE model with the lowest BPC on the validation set.

| $N_\mathrm{a}$ | $r_\mathrm{a}$(%) | $M$ | $D$ | Epoch | $M$ | $D/N$ | $\mu$ | $E$ | $K$ | $D_\mathrm{e}$ | $D_\mathrm{se}$ | $\eta$ | $B$ | # Iters | BPC |
|---|---|---|---|---|---|---|---|---|---|---|---|---|---|---|---|
| 7.30e8 | 11.19 | 5.59e9 | 5.11e11 | 7.52 | 2.86e21 | 78.47 | 21.00 | 82 | 2 | 512 | 1024 | 7.23e-4 | 1344 | 185816 | 0.4656 |
| 8.74e8 | 13.41 | 6.46e9 | 4.43e11 | 6.51 | 2.86e21 | 67.93 | 21.00 | 81 | 3 | 512 | 1536 | 6.92e-4 | 1232 | 175482 | 0.4618 |
| 1.02e9 | 15.63 | 7.33e9 | 3.90e11 | 5.74 | 2.86e21 | 59.89 | 21.00 | 80 | 4 | 512 | 2048 | 6.64e-4 | 1152 | 165447 | 0.4601 |
| 1.31e9 | 20.07 | 9.07e9 | 3.16e11 | 4.65 | 2.86e21 | 48.50 | 21.00 | 78 | 6 | 512 | 3072 | 6.23e-4 | 1024 | 150729 | **0.4590** |
| 1.71e9 | 26.18 | 1.15e10 | 2.50e11 | 3.67 | 2.86e21 | 38.32 | 21.00 | 86 | 10 | 448 | 4480 | 5.80e-4 | 960 | 127035 | 0.4597 |
| 1.96e9 | 30.07 | 1.30e10 | 2.21e11 | 3.24 | 2.86e21 | 33.83 | 21.00 | 84 | 12 | 448 | 5376 | 5.57e-4 | 792 | 135956 | 0.4603 |

Table 15: Experimental settings and results of strict data reuse ($\hat{D} = 65$B) for MoE models with $N = 3.29$B with fixed $C$. Hyperparameters shared by all experiments: $L = 24, S = 2048, D_\mathrm{m} = 1408, D_\mathrm{ffn} = 3904, H = 11, D_\mathrm{h} = 128$. The green row corresponds to the MoE model with the lowest BPC on the validation set.

| $N_\mathrm{a}$ | $r_\mathrm{a}$(%) | $D$ | Epoch | $M$ | $C$ | $D/N$ | $E$ | $K$ | $D_\mathrm{e}$ | $D_\mathrm{se}$ | $\eta$ | $B$ | # Iters | BPC |
|---|---|---|---|---|---|---|---|---|---|---|---|---|---|---|
| 2.78e8 | 8.46 | 5.41e11 | 8.33 | 2.51e9 | 1.36e21 | 164.62 | 89 | 1 | 352 | 352 | 3.24e-3 | 1728 | 152927 | 0.4916 |
| 3.47e8 | 10.54 | 4.68e11 | 7.20 | 2.92e9 | 1.36e21 | 142.36 | 88 | 2 | 352 | 704 | 3.10e-3 | 1600 | 142822 | 0.4841 |
| 4.83e8 | 14.70 | 3.75e11 | 5.78 | 3.74e9 | 1.40e21 | 114.19 | 86 | 4 | 352 | 1408 | 2.89e-3 | 1344 | 136378 | 0.4774 |
| 6.20e8 | 18.83 | 3.09e11 | 4.76 | 4.56e9 | 1.41e21 | 94.06 | 84 | 6 | 352 | 2112 | 2.73e-3 | 1280 | 117950 | **0.4757** |
| 8.93e8 | 27.12 | 2.23e11 | 3.43 | 6.19e9 | 1.38e21 | 67.74 | 8 | 1 | 3528 | 3528 | 2.465e-3 | 1024 | 106335 | 0.4794 |
| 1.14e9 | 34.75 | 1.80e11 | 2.77 | 7.69e9 | 1.39e21 | 54.85 | 84 | 15 | 320 | 4800 | 2.31e-3 | 896 | 98256 | 0.4786 |
| 1.44e9 | 43.77 | 1.47e11 | 2.26 | 9.48e9 | 1.39e21 | 44.52 | 8 | 2 | 3176 | 6352 | 2.169e-3 | 768 | 93460 | 0.4799 |
| 1.66e9 | 50.63 | 1.29e11 | 1.98 | 1.08e10 | 1.39e21 | 39.09 | 84 | 26 | 288 | 7488 | 2.08e-3 | 704 | 89125 | 0.4830 |
| 1.89e9 | 57.40 | 1.14e11 | 1.75 | 1.22e10 | 1.39e21 | 34.61 | 8 | 3 | 2888 | 8664 | 2.006e-3 | 672 | 82833 | 0.4823 |

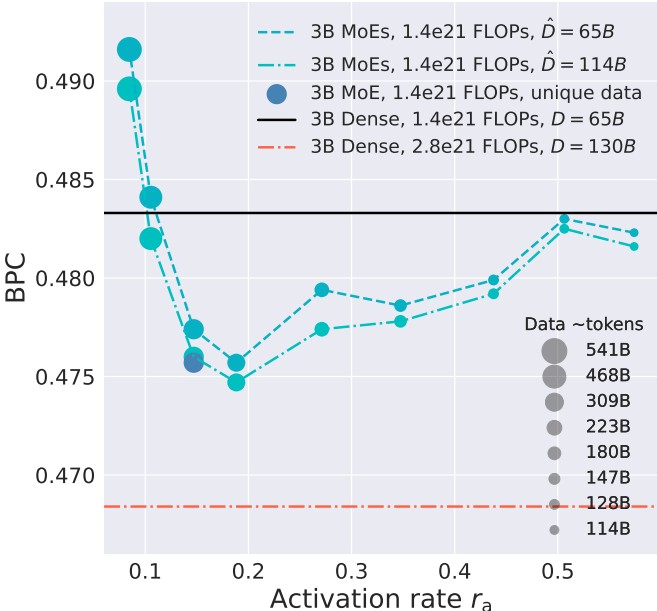

Figure 5: Performance of $N \approx 3$B models trained with varying data sizes $D$ and activation rate $r_a$. The optimal activation rate, $r_a^{**} = 20\%$, aligns with the findings for the 2B models (Figure 1). Additionally, compared to training on the unique dataset, the data reuse scheme shows only a slight performance reduction. To save computational costs, only one model trained on unique data is included for reference.

Table 16: Experimental settings and results of data reuse ($\hat{D} = 114$B) for MoE models with $N = 3.29$B with fixed $C$. Hyperparameters shared by all experiments: $L = 24, S = 2048, D_m = 1408, D_{ffn} = 3904, H = 11, D_h = 128$. The green row corresponds to the MoE model with the lowest BPC on the validation set.

| $N_a$ | $r_a(\%)$ | $D$ | Epoch | $M$ | $C$ | $D/N$ | $E$ | $K$ | $D_e$ | $D_{se}$ | $\eta$ | $B$ | # Iters | BPC |
|---|---|---|---|---|---|---|---|---|---|---|---|---|---|---|
| 2.78e8 | 8.46 | 5.41e11 | 4.75 | 2.51e9 | 1.36e21 | 164.62 | 89 | 1 | 352 | 352 | 3.24e-3 | 1728 | 152927 | 0.4896 |
| 3.47e8 | 10.54 | 4.68e11 | 4.11 | 2.92e9 | 1.36e21 | 142.36 | 88 | 2 | 352 | 704 | 3.10e-3 | 1600 | 142822 | 0.4820 |
| 4.83e8 | 14.70 | 3.75e11 | 3.29 | 3.74e9 | 1.40e21 | 114.19 | 86 | 4 | 352 | 1408 | 2.89e-3 | 1344 | 136378 | 0.4760 |
| 6.20e8 | 18.83 | 3.09e11 | 2.71 | 4.56e9 | 1.41e21 | 93.93 | 84 | 6 | 352 | 2112 | 2.73e-3 | 1280 | 117950 | **0.4747** |
| 8.93e8 | 27.12 | 2.23e11 | 1.96 | 6.19e9 | 1.38e21 | 67.74 | 8 | 1 | 3528 | 3528 | 2.465e-3 | 1024 | 106335 | 0.4774 |
| 1.14e9 | 34.75 | 1.80e11 | 1.58 | 7.69e9 | 1.39e21 | 54.85 | 84 | 15 | 320 | 4800 | 2.31e-3 | 896 | 98256 | 0.4778 |
| 1.44e9 | 43.77 | 1.47e11 | 1.29 | 9.48e9 | 1.39e21 | 44.52 | 8 | 2 | 3176 | 6352 | 2.169e-3 | 768 | 93460 | 0.4792 |
| 1.66e9 | 50.63 | 1.29e11 | 1.13 | 1.08e10 | 1.39e21 | 39.09 | 84 | 26 | 288 | 7488 | 2.08e-3 | 720 | 87144 | 0.4825 |
| 1.89e9 | 57.40 | 1.14e11 | 1.00 | 1.22e10 | 1.39e21 | 34.61 | 8 | 3 | 2888 | 8664 | 2.006e-3 | 672 | 82833 | 0.4816 |

Table 17: Experimental settings and results of loose data reuse for MoE models with $N = 6.52$B with fixed $C$. Hyperparameters shared by all experiments: $L = 24, S = 2048, D_m = 2048, D_{ffn} = 5464, H = 16, D_h = 128$. The green row corresponds to the MoE model with the lowest BPC on the validation set.

| $N_a$ | $r_a(\%)$ | $\hat{D}$ | $M$ | $C$ | $D/N$ | $E$ | $K$ | $D_e$ | $D_{se}$ | $\eta$ | $B$ | # Iters | BPC |
|---|---|---|---|---|---|---|---|---|---|---|---|---|---|
| 7.30e8 | 11.19 | 2.56e11 | 5.59e9 | 2.86e21 | 78.47 | 82 | 2 | 512 | 1024 | 7.23e-4 | 1344 | 185816 | 0.4591 |
| 8.74e8 | 13.41 | 2.21e11 | 6.46e9 | 2.86e21 | 67.93 | 81 | 3 | 512 | 1536 | 6.92e-4 | 1232 | 175482 | 0.4557 |
| 1.02e9 | 15.63 | 1.95e11 | 7.33e9 | 2.86e21 | 59.89 | 80 | 4 | 512 | 2048 | 6.64e-4 | 1152 | 165447 | 0.4550 |
| 1.31e9 | 20.07 | 1.58e11 | 9.07e9 | 2.87e21 | 48.50 | 78 | 6 | 512 | 3072 | 6.23e-4 | 1024 | 150729 | **0.4549** |
| 1.71e9 | 26.18 | 1.25e11 | 1.15e10 | 2.86e21 | 38.32 | 86 | 10 | 448 | 4480 | 5.80e-4 | 960 | 127035 | 0.4570 |
| 1.96e9 | 30.07 | 1.10e11 | 1.30e10 | 2.86e21 | 33.83 | 84 | 12 | 448 | 5376 | 5.57e-4 | 792 | 135956 | 0.4583 |

## F   THE USE OF LARGE LANGUAGE MODELS: AN EXPLANATION

Only a small fraction of complex paragraphs are written with the assistance and modification of ChatGPT. For instance, we provide the prompt: "I am writing an academic conference paper in the field of computer science. Please help me polish the wording of this paragraph, organize the sentences, and express them in a more academic way." After that the outputs are rigorously reviewed to ensure accuracy and appropriateness.

