# MIXTURE-OF-EXPERTS CAN SURPASS DENSE LLMS UNDER STRICTLY EQUAL RESOURCE

## ABSTRACT

Mixture-of-Experts (MoE) language models dramatically expand model capacity and achieve remarkable performance without increasing per-token compute. However, can MoEs *surpass* dense architectures under strictly equal resource constraints — that is, when the total parameter count, training compute, and data budget are identical? This question remains under-explored despite its significant practical value and potential. In this paper, we propose a novel perspective and methodological framework to study this question thoroughly. First, we comprehensively investigate the architecture of MoEs and achieve an optimal model design that maximizes the performance. Based on this, we subsequently find that an MoE model with activation rate in an optimal region is able to outperform its dense counterpart under the same total parameter, training compute and data resource. More importantly, this optimal region remains consistent across different model sizes. Although additional amount of data turns out to be a trade-off for enhanced performance, we show that this can be resolved via reusing data. We validate our findings through extensive experiments, training nearly 200 language models at 2B scale and over 50 at 7B scale, cumulatively processing 50 trillion tokens. All code and models will be released publicly.

## 1 INTRODUCTION

In recent years, Large Language Models (LLMs) based on the Transformer architecture (Vaswani, 2017) have achieved impressive results on a broad range of NLP tasks (Radford, 2018;

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

Table 4 in Appendix D presents the experimental settings and results. The conclusions are as follows: 1) `1dense`+SE performs the best, possibly because the dense layer contributes to more stable training. 2) The ratio of shared expert size to total expert size $D_\mathrm{se}/(D_\mathrm{se} + KD_\mathrm{e})$ has minimal impact on model performance. Therefore, we continue using the `1dense`+SE configuration and set $D_\mathrm{se} = KD_\mathrm{e}$.

**Gate score normalization.** The results of normalizing gate scores of chosen experts are in Table 5 in Appendix D. Although the addition of normalization does not show an obvious difference in performance loss, it tends to reduce the average balance loss $\bar{\mathcal{L}}_\mathrm{balance}$. Since normalization requires $K > 1$ to avoid zero gradient, we opt not to normalize given that some of our experiments have $K=1$.

**Top-K setting.** In this part, we discuss the allocation of parameters within MoE layers, focusing on the top-$K$ setting. Expert granularity is adjusted by varying $K$ and $D_\mathrm{e}$ while keeping their product constant. We conduct three groups of experiments with various $r_\mathrm{a}$ and the results are given in Table 6 in Appendix D. Note that within each experiment group, the product $K \cdot D_e$ is not strictly maintained due to compatibility with other hyperparameters. We observe that larger $K$ generally hurts performance, except in the first group where $K=1$. We attribute this exception to the $K=1$ setting. Therefore, we prevent using large $K$ and setting $K=1$ whenever possible.

**Model shape ratios.** As discussed in § 3.2, the shape hyperparameters include three ratios: $\zeta$, $\alpha$, and $\beta$. We set $\alpha = 2.77$ (Touvron et al., 2023b) and explore the optimal $\zeta$ and $\mu$, from which $\beta$ can be derived. As illustrated in Fig. 4 in Appendix D, although performance fluctuates wildly given a

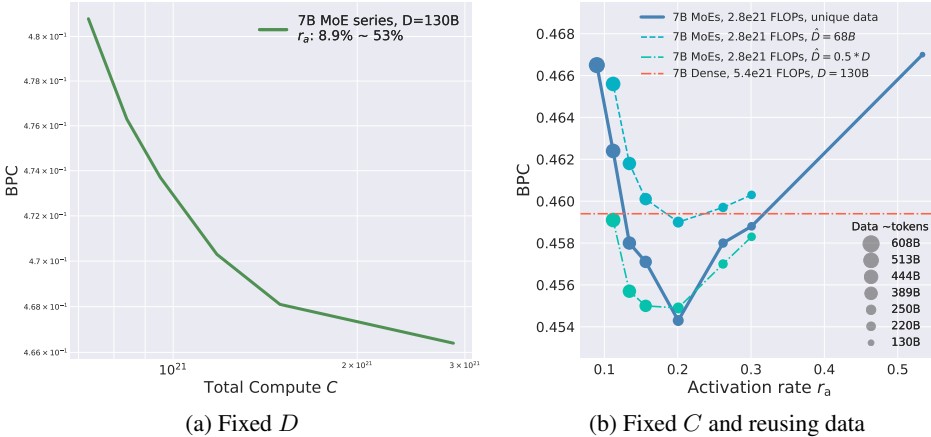

(a) Fixed $D$          (b) Fixed $C$ and reusing data

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

^*$.** Surprisingly, the loose scheme (green dashed line) often outperforms training with unique dataset. Similar results are observed on certain downstream tasks (§ 7).

**Discussion.** Prior works have explored the effectiveness of multi-epoch training for dense and MoE models. Muennighoff et al. (2023) developed a scaling law that accounts for the number of repeated tokens and found negligible loss for repeating up to 4 epochs compared to training on unique data, whereas Hernandez et al. (2022) showed degradation for dense models. Xue et al. (2023) noticed no significant gain for MoEs with repeated training when high-quality data is insufficient. Here we performed multi-epoch training on partial subsets of the unique dataset under the optimal AR setting, and found that 2-epoch data reuse boosts MoE performance over unique data, and the performance

degradation is minimal when the number of epochs further increases. We hypothesize that MoE routers benefit from additional epochs, potentially leading to more refined routing decisions and thereby mitigate the negative impact observed in dense models.

## 7 ANALYSIS OF DOWNSTREAM PERFORMANCE

To assess whether the optimal ARs generalize to downstream tasks, we conduct SFT on our 7B pre-trained models (trained w/ and w/o strict data reuse) and evaluate both the pre-trained models and SFT-ed models on a total number of 29 benchmarks (see Fig. 3 and Tab. 2), including categories such as `reasoning` and `knowledge`. The comprehensive list of benchmarks can be found in Appendix C. For all SFT trainings, we use a fixed data size $D$, and thus varying $C$ across models with different $r_a$. The 7B dense model trained with *twice* the compute is included for comparison.

**MoE *vs.* Dense at $r_a = r_a^{**}$.** For both PT and SFT models, MoEs outperform their dense equivalents across all benchmark types when $r_a = 20\%$. This result aligns with upstream findings that the optimal activation rate ($r_a^{**}$) is 20%, highlighting the very possible *universality of the optimal AR point across different training phases and data domains*. Furthermore, $r_a^{**}$ remains unchanged during SFT, even with varying $C$, suggesting that the SFT data size may have an upper limit for performance improvement, provided the PT model is adequately trained. Additionally, the `average` performance at $r_a \neq r_a^{**}$ is consistently inferior to that of dense models, underscoring the critical role of the optimal AR point.

**MoE *vs.* Dense at $r_a < r_a^{**}$.** Dense models outperform MoEs across all domains for PT models. After SFT, MoEs overtake dense models on `comprehensive` and `knowledge` tasks. However, a notable performance gap remains in `math`, highlighting the PT stage's importance for math ability.

**MoE *vs.* Dense at $r_a > r_a^{**}$.** Compared to the dense models, MoEs perform better on `knowledge` but worse on `reasoning` for PT models, and usually slightly underperform after SFT.

**Sparser *vs.* Denser.** For PT models, denser MoEs ($r_a > r_a^{**}$) outperform or match the performance of sparser MoEs ($r_a < r_a^{**}$) across all domains, consistent with Figure 2b, regardless of data reuse. When training on unique data, sparser MoEs perform better on `knowledge`. Notably, denser MoE performance significantly degrades with data reuse, especially for SFT.

**Impact of data reuse.** For both PT and SFT models, data reuse has little impact on `reasoning` but causes significant degradation in `knowledge` performance. Surprisingly, at $r_a = r_a^{**}$, the SFT-ed MoEs trained with data reuse outperform both MoEs and dense models trained on unique data. This implies that *a model can master reasoning skills (rather than merely memorizing information (Hu et al., 2024)) with a relatively small dataset (Muennighoff et al., 2025; Wang et al., 2025) and further enhance its capabilities through multiple training epochs*.

## 8 CONCLUSION AND FUTURE WORKS

In this paper, we propose a three-step experimental methodology to investigate whether MoEs can surpass their dense counterparts under the same constraints on total parameters, compute, and data. By optimizing the architecture, identifying the optimal activation rate region, and reusing data, we arrive at a positive answer to this question. Future work will explore how optimal activation rates enhance model capabilities and whether similar conclusions hold for other training methods like upcycling (Komatsuzaki et al., 2022) and MoEfication (Zhang et al., 2021). We hope this work offers valuable insights for the architectural design of next-generation models.

### LIMITATIONS

The limitations of this work include: 1) Hindering by the high computational cost, we did not train models larger than 7B. 2) As described in § 4, we focus mainly on the impact of several main components of MoEs, but fix the rest to narrow the scale of experiments. 3) Exploration of other elements can provide further comprehensive guidance for the architectural design of MoEs.

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

# APPENDIX

## A    BACKGROUND: MIXTURE-OF-EXPERTS

The MoE architecture primarily consists of a gate and several experts. Typically, the gate $g(\cdot)$ is composed of a linear layer $W_g$ followed by a Softmax and a Top-K operation, and the experts $E_{i=1,...N}$ follow the standard FFN structure. The computation of an MoE block can then be formulated as follows:

$$y = \sum_{i=1}^{N} g_i(x) \cdot E_i(x), \tag{11}$$

$$g_i(x) = \begin{cases} s_i & \text{if } s_i \in \text{TopK}(s;K), \\ 0 & \text{otherwise}, \end{cases} \tag{12}$$

where $s_i = \text{Softmax}_i(W_g x)$ denotes the gate score for the $i$-th expert.

## B    EXTENDED ANALYSIS ON RELATED WORK

We notice a concurrent work (Abnar et al., 2025) studied scaling law for optimal MoE sparsity. We highlight the differences between our work and theirs as follows:

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

Table 6: Experimental settings and results of top-K setting. Hyperparameters shared by all experiments: $\text{Scheme} = \texttt{1dense}, L = 16, D_{\text{m}} = 1408, D_{\text{ffn}} = 3904, H = 11, D_{\text{h}} = 128, \text{Norm} = \text{False}$.

| $N$ | $N_{\text{a}}$ | $r_{\text{a}}$ (%) | $M$ | $E$ | $K$ | $D_{\text{e}}$ | $D_{\text{se}}$ | $\mathcal{L}$ |
|---|---|---|---|---|---|---|---|---|
| 2.15B | 591M | 27.47 | 8.00e9 | 8 | 1 | 3528 | 3528 | 2.0470 |
| 2.15B | 591M | 27.40 | 8.00e9 | 88 | 11 | 320 | 3520 | **2.0338** |
| 2.15B | 949M | 44.00 | 1.01e10 | 8 | 2 | 3176 | 6352 | **1.9996** |
| 2.15B | 948M | 44.05 | 1.01e10 | 88 | 22 | 288 | 6336 | 2.0266 |
| 2.15B | 1.24B | 57.57 | 1.19e10 | 8 | 3 | 2888 | 8664 | **2.0156** |
| 2.11B | 1.22B | 57.68 | 1.18e10 | 88 | 33 | 256 | 8448 | 2.0235 |

Table 7: Experimental settings and results of model shape ratios. Hyperparameters shared by all experiments: $\text{Scheme} = \texttt{1dense}, S = 16384, D_{\text{h}} = 128$.

| $N$ | $N_{\text{a}}$ | $D_{\text{m}}$ | $D_{\text{ffn}}$ | $L$ | $H$ | $E$ | $K$ | $D_{\text{e}}$ | $D_{\text{