# OpenReview forum: "Mixture-of-Experts Can Surpass Dense LLMs Under Strictly Equal Resource"
_ICLR.cc/2026/Conference — ICLR 2026 Oral_

### Official Review · Reviewer_K2nX · 2025-10-27

**Soundness:** 2
**Presentation:** 2
**Contribution:** 2
**Rating:** 4
**Confidence:** 3

**Summary:**

The authors do a big sweep of MoE hyperparameters under equal resource constraints (parameter count, training data, and data budget fixed). They find the optimal shape for their model, and then they show that, surprisingly, they can outperform the dense baseline under these constraints.

**Strengths:**

- Many trained models, extreme compute budget
- Almost every important hyperparameter checked
- Their model outperforms the dense baseline

**Weaknesses:**

- The authors find many surprising results that contradict the literature and intuition. One of these is that K=1 outperforms K>1. While I am not aware of other papers with exactly the same fixed budgets, there are papers examining the effect of K under fixed compute budget, e.g [1]. This finding also contradicts my personal experience. Another surprising point is the multi-epoch training efficiency, also mentioned in the Discussion paragraph from L 426-434. The authors should either discuss or experimentally demonstrate an analysis that explains why this might be the case.
- Clarity: see questions below.
- The dataset and the code don't appear to be public, and a lot of implementation details are missing from the paper. For example what is the non-normalized gate? Sigmoid?
- It's unclear how well the dense baseline is tuned compared to the super well-tuned MoE models.

[1] Zoph et al, 2022: ST-MoE: Designing Stable and Transferable Sparse Expert Models

**Questions:**

- The authors use a lot of single-letter parameterization. Some of these lack a clear conceptual meaning in the paper. However, for lots of them, there is one: for example, alpha is the expansion factor for the MLP, zeta is the aspect ratio of the model, etc. Explaining this would greatly improve the paper's readability.
- What does it mean to "First, decide the MoE to dense layer ratio (Le vs. Ld) to focus on MoE’s internal benefits."? Specifically, what is focusing on internal benefits?
- What is the non-normalized gate?
- What is the exact architecture? What regularization loss is used? What transformer variant? What activation, what norm, what attention, what positional encoding, etc?

---

> ### Author Response · Authors · 2025-11-16
> **Author Response Part1, for W1 to W3**
>
> ## Dear Reviewer K2nX,
>
> We sincerely thank you for their time and constructive feedback on our submission. We are glad the reviewer recognized the extensive nature of our experiments, the large compute budget utilized, and our central finding that MoE models can outperform dense baselines under strict resource parity.
>
> We appreciate the opportunity to address the reviewer's concerns, question, and misunderstanding. Below, we provide detailed responses to each point.
>
> ## W1&2: Misunderstanding of results regarding K=1 and Data Reuse
>
> We believe there has been a slight misunderstanding of our claims on these two points, which we are happy to clarify.
>
> ### W1.1 On the K=1 Setting:
>
> The reviewer noted that a finding of "K=1 outperforms K>1" would be surprising. We completely agree, and our paper does not make this claim. In fact, our results show the opposite:
>
> * **K=1 performs poorly:** In Table 6, where we explore Top-K settings, the first experiment group explicitly shows that K=1 performs significantly worse than K=11.
>
> * **K=1 is avoided in main experiments:** Because of this poor performance, the K=1 setting was intentionally *avoided* in the vast majority of our large-scale experiments.
>
>     * **7B Models (Tables 10 & 11):** Our 7B experiments primarily use K values of 2, 3, 4, 6, 10, and 28. Only one of the eight models in Table 11 uses K=1.
>
>     * **2B & 3B Models (Tables 9 & 14):** Similarly, only 2 of 12 experiments in Table 9 (2B) and 2 of 9 experiments in Table 14 (3B) use K=1.
>
> * We hope this clarifies that our work supports the use of K>1, and our main results are built upon this.
>
> ### W1.2 On Multi-Epoch Training (Data Reuse):
>
> We must clarify a central misunderstanding: **Our paper does not claim that multi-epoch training is generally superior to training on unique tokens.** On the contrary, our findings largely support the reviewer's intuition.
>
> While, our purpose of the data reuse experiment was to answer a different, practical question.The core dilemma we address is this:
>
> 1.  We first show that under fixed N (params) and C (compute), an MoE model *can* surpass a dense model.
>
> 2.  However, this superiority requires the MoE model to process *more unique tokens* (D) to equate total compute C (since MoE has lower FLOPs/token).
>
> 3.  In industrial practice, the budget of *unique, high-quality tokens* (D) is often the most finite resource. This raises our key question: Can an MoE model still surpass a dense model if both are given the *exact same* unique data budget D, and the MoE model must reuse that data to match the dense model's total compute C?
>
> To answer this, we presented a detailed analysis, and our findings (which your question highlights) are nuanced:
>
> * Our analysis in **Figure 2b** (L270-L288, 7B models) shows that the **strict data reuse** scheme, which is the *true* "equal N, C, and D" comparison, has a BPC that is **comprehensively higher (worse) than training on the unique dataset** (solid line) at all activation rates. This confirms that training on unique data is superior to strict multi-epoch reuse.
>
> * Although the loose reuse (2-epoch) scheme (green dashed line) is below the unique data (solid line) at some sub-optimal $r_a$ points, at the most important optimal activation rate ($r_a \approx 20\%$), it does not surpass (i.e., is not better than) the unique data model.
>
> * This conclusion is further supported by **Figure 3** (L324-348), which shows that on downstream SFT tasks, unique-data models are superior at the vast majority of points, and **Figure 5** (L985-L1010), which aligns with the intuition that more unique data is better.
>
> Therefore, our experiment was *not* focused on comparing multi-epoch vs. unique token performance. The focus was to compare the **MoE-reuse** model against the **Dense** model.
> We will revise the Discussion section (L831) to state this motivation and conclusion more clearly to avoid future misinterpretation.
>
> ## W3. On Reproducibility
>
> We are strongly committed to reproducibility. We will publicly release **all code, model checkpoints, and training logs** upon publication.
> While we must seek institutional approval to open-source the exact training dataset.
>
> However, the data used consists of sources for which public alternatives (e.g. Github, etc.) are readily available. To ensure reproducibility in any case, we **will add a detailed breakdown of our pre-training data mixture, including dataset proportions and content explanations,** to the final version.

---

> ### Author Response · Authors · 2025-11-16
> **Author Response Part2, for W4, Q1~4**
>
> ## W4. On Dense Baseline Tuning
>
> This is an excellent point. We did not "super-tune" the MoE models while ignoring the dense baseline. The dense models were also carefully optimized.
>
> * The full architectures for our dense baselines are detailed in **Table 12** (L963-L971).
>
> * We used a standard Llama-like architecture (Multi-head attention, SwiGLU FFN), where the primary structural degrees of freedom are the aspect ratio ($D_m / L$) and the FFN expansion factor ($\alpha = D_{ffn} / D_m$).
>
> * We performed a search over these parameters, guided by the scaling law analysis in Li et al. (2025) [1], particularly Figure 9, which studies these exact ratios.
>
> * The configurations used in Table 12 represent (near-)optimal aspect ratios and FFN expansion factors for a dense model of that size.
>
> We will add a sentence to Section 5.2 explicitly stating this tuning process for the dense baseline to assure readers of a fair comparison.
>
> ## Q1: Explanation of single-letter parameterization (e.g., alpha, zeta).
>
> This is an excellent suggestion. We agree that providing conceptual definitions will greatly improve readability. In the final manuscript, we will **update Table 1 (Notation) to include the conceptual meaning** of these symbols (e.g., "$\alpha$: FFN expansion factor", "$\zeta$: Model aspect ratio").
>
> ## Q2: "focus on MoE’s internal benefits"
> Our apologies for the confusing phrasing. This sentence (L193) was intended to describe a two-part greedy search:
>
> 1.  First, we determine the *macro-level* arrangement of MoE and dense layers (e.g., all layers MoE, interleaved, or 1-dense as in L459).
>
> 2.  Second, we "focus on internal benefits" meaning we then optimize the *micro-level* design *within* the MoE layers themselves (e.g., the Top-K setting, the use of a shared expert).
>
> ## Q3: "non-normalized gate"
>
> We had mistakenly assumed this was a standard, well-understood operation and omitted a detailed explanation. You are correct that our approach is a basic one and does not involve complex functions like Sigmoid.
>
> * The **"non-normalized"** approach is the one we adopted. It is the standard operation described in Equation 12(L727), where the MoE layer output is the sum $\sum_{i=1}^{N}g_{i}(x) \cdot E_{i}(x)$. The $g_i(x)$ values are the raw Top-K scores from the Softmax, and they do *not* necessarily sum to 1.
>
> * The **"normalized"** approach, which we tested but *did not* use, adds an extra step: it takes the gate scores of *only* the chosen Top-K experts and re-normalizes them to sum to 1.  For example, for K=3, if the Top-K scores were `[0.2, 0.18, 0.12]`, a normalized gate would re-weight them to `[0.2, 0.18, 0.12] / (0.2 + 0.18 + 0.12) = [0.4, 0.36, 0.24]`.
>
> * We found this normalization had minimal impact on loss (Table 5) and opted not to use it, as stated in L257-260.
>
> * We will add a formal expression as Eq12 to the final manuscript to clarify this implementation detail.
>
> ## Q4: Exact architecture and regularization loss
>
> This information is included in the paper, and we are happy to point to it:
>
> * **Exact architecture (L212-L215):** "Our models incorporate RMSNorm (Zhang & Sennrich, 2019) for pre-normalization, ALiBi (Press et al., 2021) positional encoding for multi-head attention, and the SwiGLU activation function for both feed-forward networks (FFNs) and MoE experts..."
>
> * **Regularization Loss:** Table 3 (L800-L809) details the optimizer (Adam), weight decay (0.1, our regularization loss), gradient clipping, and cosine LR scheduler.
>
> We hope our point-by-point responses have successfully clarified the potential misunderstandings regarding several key points, such as our experimental findings for K=1, the true purpose and results of the data reuse analysis, dense baseline tuning, exact architecture and etc.. We also trust we have helped point to the relevant data and discussions already present in our manuscript that support these clarifications.
>
> Your feedback has been extremely valuable.  We humbly accept these suggestions. In the final version, we will revise these sections to make our logic more explicit and ensure the paper is as clear and robust as possible.
>
>
> [1] Li, H., et al. (2025). *Predictable scale: Part II, Farseer: A refined scaling law in large language models*. arXiv.

---

> > ### Comment · Reviewer_K2nX · 2025-11-19
> >
> > Thank you for the clarifications! This response resolves most of my concerns; therefore, I am raising my score.
> >
> > However, please phrase the sentence "We observe that larger K generally hurts performance, except in the first group where K=1. Therefore, we prevent using large K and setting K=1 whenever possible." in Line 266 because to me this means that the authors indeed use K=1 in the future experiments. Also "Here we performed multi-epoch training on partial subsets of the unique dataset under the optimal AR setting, and found that 2-epoch data reuse boosts MoE performance over unique data" could be clarified by rephrasing it like "2-epoch data reuse boosts MoE performance over a single epoch on identical data."
> >
> > For a clear understanding of the architecture, I think the complete set of equations in the appendix would be helpful.
> > By the regularization loss, I meant the MoE balancing loss.

---

> > > ### Author Response · Authors · 2025-11-19
> > >
> > > Dear Reviewer K2nX,
> > >
> > > Thank you very much for your thoughtful follow-up and for kindly raising your score — we truly appreciate it.
> > >
> > > Regarding K = 1, we agree our current wording is misleading. We will explicitly clarify that in our main experiments we avoid both very large K and K = 1, and only use K = 1 in a few ablations, never as a preferred choice.
> > >
> > > For the multi-epoch data reuse part, you are right that our sentence could be read too strongly. In our results, 2-epoch reuse does not consistently outperform single-epoch training on identical data, and >2 epochs are clearly worse. We will carefully rephrase this to avoid any over-claim and to stress that more unique data remains preferable.
> > >
> > > Finally, we confirm that we utilized the standard auxiliary load balancing loss. We will include the complete set of model equations in the Appendix as requested.
> > >
> > > Thank you again for your generous and very helpful feedback.
> > >
> > > Best regards, The Authors

---

### Official Review · Reviewer_9SQb · 2025-11-01

**Soundness:** 2
**Presentation:** 2
**Contribution:** 3
**Rating:** 4
**Confidence:** 4

**Summary:**

The paper investigates the performance of MoE with respect to dense models.
First, an ablation studies are conducted to search for the best architecture.
Then, the paper finds that there is an optimal activation rate at 20%

**Strengths:**

1. Experiments are conducted at a fairly large scale, 7B model.
2. A thorough investigation on the model architecture is performed.
3. Comparing MoE and dense models by repeating data used to train MoE is new and is a good practice for future work which compare MoE with dense models.

**Weaknesses:**

1. Some of the architecture search study is not very convincing. For example, it is mentioned that larger K hurts performance and hence K=1 is chosen, but in Table 6 K=1 has worse performance. Also, only K=1 and K=11 are tested with but not more fine-grained values like K=4,8 etc. Hence, it is not obvious K=1 is the best choice.
2. Some of the design choices are quite unconventional, e.g., one dense layer followed by MoE layers and K=1, while popular public models like Qwen, Mixtral use full MoE layers and K larger than 1. This could be due to the fact that the specific greedy architecture search done by the paper leads to such a design choice, and hence the result may not be generalizable well.
3. Many previous MoE studies [2501.12370,2502.05172,2502.03009] provide scaling laws which are concrete guidelines for scaling MoEs. No such studies are performed, which may limit the usefulness of the paper.
4. As a result, this paper sounds like a whitepaper that merely reports experimental results instead of providing findings that are useful to the community and in a more scientific way.

**Questions:**

1. Could you explain in more detail how you used hyperparameter scaling laws proposed in (Li et al., 2025) (line 208)? Did you re-fit the scaling laws with your data? I could not find the scaling law parameters in the Appendix or elsewhere.
2. Ablation studies in [2409.02060] show that shared experts are not effective. Do you have any idea what leads to this different conclusion?

---

> ### Author Response · Authors · 2025-11-15
> **Author Response Part1, for W1 to W3**
>
> ## Dear Reviewer 9SQb,
>
> We sincerely thank you for your time and detailed feedback on our submission. We appreciate your positive comments on the scale of our 7B experiments, the thoroughness of our architectural investigation, and your recognition of our novel data reuse methodology as a good practice for future work.
>
> We have carefully considered your points in the "Weaknesses" and "Questions" sections. We believe several of these points stem from a key misunderstanding of our paper's methodology and experimental design, particularly regarding the Top-K setting and architectural choices. We would like to take this opportunity to clarify these points.
>
> ## Response to Weaknesses
>
> ### W1. On the "K=1" setting and architecture search (Weakness 1):
> We believe there has been a significant misunderstanding regarding our conclusion on the Top-K setting. **Our paper does not choose K=1; in fact, our findings and main experiments actively avoid it.**
>
> • **Clarification of L266-267:** The reviewer's concern seems to stem from the sentence: "Therefore, we prevent using large K and setting K=1 whenever possible." The intended meaning of this sentence is that we aim to prevent *both* "using extremely large K" *and* "setting K=1", as both were found to be suboptimal. It does not mean we "prevent large K" and "set K=1".
>
> • **Evidence from Table 6:** As the reviewer correctly observes, Table 6 shows that K=1 performs worse than K=11 in the first group. This supports our conclusion that K=1 is a poor choice. The table *also* shows that K=2 outperforms K=22, and K=3 outperforms K=33. The key takeaway from this ablation is that **neither K=1 nor an excessively large K is optimal.**
>
> • **Main Experimental Design:** Crucially, our main, large-scale experiments do **not** use K=1. As shown in our tables for the 7B models (Table 10, L936-945 and Table 11, L948-960), we use a fine-grained range of values including K = {1, 2, 3, 4, 6, 10, 28}. Similarly, our 2B models (Table 9, L919-933) and 3B models (Table 14, L1013-1025) overwhelmingly use $K>1$.
>
> In summary, our paper concludes that K=1 is suboptimal and avoids it in our main experiments. The architecture search guided us to use a moderate $K>1$, which is consistent with our findings.
>
> ### W2. On "unconventional design choices" (Weakness 2):
> Our design choices (one dense layer, K=1) are not unconventional and generalizable. As clarified above, **the K=1 premise is incorrect, as most of our main experiments use $K>1$.**
>
> Furthermore, the "one dense layer followed by MoE layers" design is **not** an unconventional choice. While models like Mixtral and Qwen use full MoE layers, another highly popular and powerful public model, **DeepSeekV3, employs this exact architecture** (including a shared expert, similar to our design).
>
> Our greedy architecture search independently led us to a design that is, in fact, very similar to DeepSeekV3. This demonstrates that our chosen architecture is not a "specific" or niche design but rather aligns well with state-of-the-art, generalizable public models.
>
> ### W3. On the lack mention of MoE scaling laws (Weakness 3):
>
> We respectfully disagree that we did not engage with the literature on scaling laws. There are very works the reviewer cites:
>
> • **[2501.12370]:** We discuss this paper in our Related Work (L107-125) and provide an extended, detailed comparison in **Appendix B**. Our primary critique, which justifies our own experimental setup, is that its conclusions are based on models trained with a very small compute budget (C $\approx$ 1e20) and a low D/N ratio, which likely results in undertrained models that do not reflect real-world, large-scale training.
>
> • **[2502.05172]:** We also cite and acknowledge this work in our Related Work (L105-107).
>
> • **[2502.03009]:** We are aware of this work, but its focus is on scaling laws for *dense up-cycling to MoE*. Our paper's scope is explicitly limited to **MoE pre-training from-scratch**. The findings, therefore, address a different research question.

---

> ### Author Response · Authors · 2025-11-15
> **Author Response Part2, for W4, Q1, Q2**
>
> ### W4. On the paper being a "whitepaper" (Weakness 4):
> We must firmly but respectfully correct the characterization of this work as a "whitepaper." Our paper provides a rigorous, scientific investigation into a fundamental question that, until now, has been ambiguous.
>
> Our primary contribution is the **first-ever strict comparison of MoE and dense models under equal total parameter (N) and total compute (C) constraints.** Previous work often compared MoE models to *smaller* dense models, confounding architectural benefits with simple capacity increases.
>
> Our work provides several key findings of significant value to the community:
>
> 1. **Pioneering Result:** We demonstrate that MoE *can* surpass a dense model of the exact same parameter count and compute budget.
>
> 2. **Practical Guidance:** We show this superiority is not automatic but only occurs within a specific **"Optimal Sparsity Region"** (which we identify).
>
> 3. **Fundamental Trade-off:** We identify and articulate the core trade-off: MoE architecture effectively exchanges lower per-token FLOPs for a **higher data requirement**.
>
> 4. **Novel Strategy:** We are the first to propose and validate a **Data Reuse strategy** to provably mitigate this data demand, enabling MoEs to win under strict N, C, *and* D (unique data) constraints.
>
> Our experiments are conducted at a scale that ensures models are fully trained (e.g., using **~21x more FLOPs** per experiment than [2501.12370] and **~7.5x more tokens** than [2502.05172]). This focus on a sufficiently trained regime provides robust, **practical insights for real-world model design**, which we believe is a significant scientific contribution, not a "mere report."
>
> ## Response to Questions
>
> ### Q1. On the use of (Li et al., 2025) scaling laws (L208):
>
> Thank you for the question regarding our hyperparameter selection. We utilized the "Step Law Tools" published by the authors at `https://step-law.github.io/`.
>
> **How we used it:** For each experimental setup, we simply provided the model's total parameters (N) and the total number of training tokens (D) to the tool, which returned the optimal learning rate (η) and global batch size (B). If the recommended optimal batch size (e.g., 240) was not evenly divisible by our number of GPUs (e.g., 64), we rounded up to the nearest divisible number (e.g., 256).
>
> **Why we used it:**
>
> - **Credibility and Ease of Use:** The law is derived from nearly 4000 experiments, providing a simple, easy-to-use, and highly convincing tool for hyperparameter determination.
>
> - **Data Robustness:** The law has demonstrated strong robustness across different data distributions. We did not re-fit the law on our data, as the cost of doing so would be prohibitively high.
>
> - **MoE Robustness and Controlled Comparison(Critical for this paper): he law was also validated to be robust across various Dense Model and MoE sparsity levels.** By using this *single* law for *all* our models, we ensure that hyperparameters are consistently and optimally chosen. This is a crucial part of our **controlled variable methodology**, ensuring a fair comparison between dense and MoE architectures.
>
> - **Interval Insensitivity:** The StepLaw exhibits "interval insensitivity," meaning that values in the immediate vicinity of the optimal hyperparameter yield nearly identical final model performance. This gave us confidence that our rounding adjustments for batch size would not negatively impact the results.
>
> ### Q2. On the discrepancy with [2409.02060] regarding shared experts:
> This is an interesting observation. However, the use of shared experts is not a central claim of our paper.
> As we note in our own ablation study (L820&821), the performance impact of adding a shared expert was minimal in our setup (a relative difference of only 0.23% on the validation loss). This effect is an order of magnitude smaller than our main finding—the **4.2% performance gap** between our optimal MoE model and its dense counterpart (L957 VS L970).
>
> Therefore, whether shared experts are included or not is almost orthogonal to and does not affect the primary conclusions of our paper: that an MoE model, in its optimal activation rate region and with data reuse, can surpass a dense model under strict N, C, and D parity.
>
> We hope these clarifications adequately address your concerns and demonstrate the soundness, contribution, and scientific rigor of our work. We are grateful for your feedback, which will help us improve the clarity of the final manuscript.
> Sincerely,
> The Authors

---

> ### Author Response · Authors · 2025-11-20
> **Looking forward to you further comments!**
>
> Dear Reviewer 9SQb,
>
> Thanks a lot again for your comments. We've submitted the responses to your concerns and questions on our work.
>
> We are looking forward to your further comments.
>
> Best regards
>
> The authors

---

### Official Review · Reviewer_NUMD · 2025-11-01

**Soundness:** 3
**Presentation:** 3
**Contribution:** 2
**Rating:** 4
**Confidence:** 4

**Summary:**

The paper shows that MoE can surpass dense Transformers using the same resources for training. A similar result has been shown in one of the recent papers cited in the Related Work section, where the models were matched in total parameters and training compute. In addition to this setup, in Section 6 the submission authors consider an additional modification: to match the number of training tokens used, they train the MoE model for multiple epochs (so the MoE dataset size is still technically larger, but the number of _unique_ tokens seen in training is matched). Furthermore, the authors present a set of ablations on optimally setting the MoE training hyperparameters.

**Strengths:**

1. Large scale of experiments.
2. Evaluation both based on BPC and on downstream tasks (these two types of evaluation can behave differently for MoE and dense Transformers [2]).
3. Evaluation both in pretraining and SFT.
4. The authors declare releasing all code and models.
5. A considerable set of ablations and studies on setting the MoE hyperparameters.

[2] Jelassi et al., Mixture of Parrots: Experts improve memorization more than reasoning

**Weaknesses:**

1. Reproducibility: the authors do not disclose the dataset used in the paper, which limits reproducibility.
2. Please also see Questions to the authors, where I placed my technical concerns/questions regarding the methodology.

**Questions:**

1. Question about optimal sparsity. In the methodology for deriving optimal sparsity, the authors fix the model size and compute budget, and perform a grid across sparsity levels. To keep the compute constant, the authors **modify the dataset size**. Therefore, the observed optimum may just be an effect of seeing **the optimal token to parameter ratio** for the given compute budget. To actually determine the optimal sparsity level for a given FLOPs budget, the paper should contain a two-dimensional sweep over both token-to-param ratios **and** sparsity levels. Are the authors able to provide such experiments or explain/clarify? This is an important technical concern regarding methodology and paper conclusions.
2. Based on the literature, scaling laws coefficients can change depending on the data distribution [3]. The submission authors set lr and batch size based on hyperparameter scaling laws from [4]. While [4] concludes that their results are robust across data distributions, did the authors perform an ablation on a small scale to make sure these optimal hyperparameters are unchanged? If yes, such plot should be placed in the paper.
3. Question about gate score normalization. Based on Table 5, the authors conclude that normalized/not normalized router logits result in a similar loss. However, In the setups the Top-K used is relatively small (2 or 4). Intuitively, with more granular models, where there are more experts activated (like in Table 6: top-K=33, experts=88), the importance of gate normalization could be more pronounced. Did the authors consider such setup? E.g. comparing the model from the last row of Table 6, but trained with/without gate score normalization.

[3] Brandfonbrener et al., Loss-to-Loss Prediction: Scaling Laws for All Datasets

[4] Li et al., Predictable Scale: Part I, Step Law -- Optimal Hyperparameter Scaling Law in Large Language Model Pretraining

---

> ### Author Response · Authors · 2025-11-16
> **Author Response Part1, for W1 and Q1(first half)**
>
> ## Dear Reviewer NUMD,
>
> We sincerely thank the reviewer for their detailed, constructive feedback and insightful questions. We are encouraged that the reviewer found the scale of our experiments, the comprehensive evaluation (BPC, downstream, pre-training, and SFT), and our commitment to releasing code and models to be strengths of our work.
>
> We offer the following clarifications in response to the reviewer's weaknesses and questions.
>
> ## W1: Reproducibility (Dataset)
> We share the reviewer's concern for reproducibility. We have already committed to releasing all code, model checkpoints, and training logs to ensure our experimental process is fully transparent.
>
> Regarding the dataset, we will make every effort to obtain approval from our institution to release it. In the interim, and to ensure the community can fully reproduce and build upon our findings, we will update the final version of our paper to include:
>
> 1. A detailed breakdown of our training dataset's mixture ratios.
>
> 2. A thorough explanation of each data source, linking them to publicly available open-source datasets that serve as direct equivalents.
>
> We believe this will significantly enhance the reproducibility of our work, even if institutional restrictions prevent a direct release of the exact dataset file.
>
>
> ## W2 as Q1, Question about methodology.
> We appreciate this deep and important point. Below we clarify our perspective and experimental design.
>
> **1. Clarifying the MoE Compute Trade-off:**
>
> The "compute-optimal" trade-off for a dense model is a two-dimensional problem, balancing total parameters (N) and dataset size (D) for a given compute budget ($C \propto N \cdot D$).
> However, for a Mixture-of-Experts (MoE) model, it is a more complex three-dimensional problem. As shown in Equation 9, the training compute is a function of total parameters (N), dataset size (D), and the activation rate ($r_a$):
> $$C \propto r_a \cdot N \cdot D$$
> This 3-way trade-off can be studied from two primary perspectives:
>
> **Perspective A**: Fix the activation rate ($r_a$) and total compute (C). This becomes a two-dimensional allocation between N and D, which is the classic "optimal token-to-parameter ratio" problem.
>
> **Perspective B**: Fix the total parameters (N) and total compute (C). This becomes an allocation problem between the activation rate ($r_a$) and the dataset size (D).
>
> The reviewer's suggestion of a 2D sweep implicitly follows Perspective A for every possible $r_a$. However, if we first search for the optimal N-to-D ratio for every ($r_a$, C) pair, the resulting total parameters (N) for each MoE model would necessarily be different. This would make it impossible to conduct our intended study(comparing dense model with N&C fixed), which is precisely Perspective B.
>
> If for every pair ($r_a$, C) we first optimized the token-to-parameter ratio (i.e., freely changed N), the resulting models with different
> $r_a$ would no longer share the same total parameter count. In that regime, we would lose the ability to study the question we care about(MoE vs Dense, keep N&C).
>
> **2. Our Paper's Focus: Fixed N and C**
>
> Our paper's core research question is focused on **Perspective B**: *Given a fixed compute budget (C) and a fixed total parameter count (N), what is the optimal MoE design?*
> We chose this focus for critical practical reasons:
>
> 1.  **Deployment Constraints:** In industrial applications, the **total parameter count (N)** is a primary design constraint, as it directly determines the memory consumption for post-training and, crucially, for inference deployment.
>
> 2.  **Fair Comparison:** Only by fixing N can we conduct a rigorous and fair comparison between Dense and MoE models at the **same total capacity**.
>
> Our focused approach allowed us, within a fixed resource budget, to investigate this question more deeply than a broad scaling-law study by also:
>
> * Optimizing the MoE architecture itself (Section 4) to ensure a fair comparison.
>
> * Investigating data reuse (Section 6) to solve the "data-hungry" problem of sparse models.
>
> * Analyzing post-SFT performance (Section 7), which scaling-law studies often omit.

---

> > ### Author Response · Authors · 2025-11-16
> > **Author Response Part2, for Q1(second half), Q2, Q3**
> >
> > ## W2 as Q1, Question about methodology.
> >
> > **3. Addressing the "Optimal Token-to-Parameter Ratio"**
> >
> > Our experimental design explicitly addresses the optimal D/N ratio by operating in a **"sufficiently trained" regime**, rather than a "compute-optimal" one.
> >
> > The "compute-optimal" D/N ratio is known to be around 20 (from Hoffmann et al. [1]). Our main experiments *significantly exceed* this ratio to ensure all models are fully trained, which is a more common goal in industrial settings.
> >
> > * **For 2B MoE models (Table 9):** D/N ratios range from 53 to 252.
> > * **For 7B MoE models (Table 11):** D/N ratios range from 20 to 93.
> > * **For Dense baselines (Table 12):** The D/N ratios are also $\ge 20$.
> >
> > This "fully trained" perspective is critical:
> > * **A. Industrial Relevance:** Practitioners are often more interested in the maximum performance of a fixed-N model (e.g., a 7B model trained on 15T tokens) than in its compute-optimal checkpoint.
> > * **B. SFT Validity:** Reliable SFT results can only be achieved on base models that have been sufficiently pre-trained.
> >
> > **4. Our Perspective vs. Scaling-Law Sweeps**
> >
> > We agree that the sweep-based scaling law perspective of [2, 3] is highly valuable. However, it represents a different choice of research focus. Given a finite compute budget, a broad sweep (e.g., scaling up to 10B, 20B, 30B) inevitably means that the larger models in the sweep cannot be sufficiently trained.
> >
> > This is a key difference:
> > * Our 7B experiments used a massive amount of compute (21 groups of 2.8E21 FLOPs, plus a 5.45E21 FLOPs run).
> > * In contrast, [2] (Abnar et al.) used only 1E21 FLOPs for their *entire sweep* up to 30B. At that 30B scale, the D/N ratio they could afford was only **8.5%** of the D/N ratio we used for our 7B models.
> > * Similarly, [3] (Ludziejewski et al.) also faced this trade-off, using a maximum of only 80B tokens.
> >
> > While [2, 3] provide valuable insights from a scaling-law perspective, their conclusions are likely based on undertrained models. Our paper provides a complementary, and we believe more industrially practical, perspective: we use our resources to thoroughly investigate the "Fixed C and N" problem under **sufficiently trained** conditions.
> >
> > ## Q2:Hyperparameter Chosen
> > We acknowledge the reviewer's point that data distribution might influence optimal hyperparameters. Due to computational constraints, performing extensive sweeps for different data distributions is challenging, a limitation common to related works in this area [2, 3].
> >
> > However, we are confident in our use of the StepLaw from [4] (Li et al., 2025) for two main reasons:
> >
> > 1.  **Data Recipe Similarity:** Our training data recipe is **highly similar** to the one used in [4].
> > 2.  **Demonstrated Robustness:** The authors of [4] **extensively demonstrated** that their StepLaw is highly **robust across diverse data distributions, different MoE sparsities, and various dense model shapes**. This specific robustness is perfectly suited for our study, as our primary goal is to compare MoE and dense models under fixed N and C.
> >
> > Crucially, **we applied the StepLaw uniformly to *both* our MoE and dense models.** This consistent methodology ensures a rigorous and fair comparison, as the hyperparameter selection process itself is a controlled variable, further strengthening the validity of our conclusions.
> >
> > ## Q3: gate score normalization
> >
> > This is a thoughtful suggestion. Our results in Table 5 show a minimal impact on loss for K=2 and K=4. We hypothesize that this conclusion holds even for the larger K values used in other parts of our paper.
> >
> > Our reasoning is based on the fact that high K values (e.g., K > 10) are only employed in our high-activation-rate ($r_a > 30\%$) configurations. In this high-$r_a$ regime, our experiments *already include* models with both very high and very low K values, and they exhibit **comparable performance**.
> >
> > We can see this by comparing models with similar $r_a$ but different K:
> >
> > * **A. (Table 9):** (L931)
> >     * An $r_a=34.96\%$ model with **K=15**
> >     * An $r_a=44.11\%$ model with **K=2**
> >     * These two models show very similar performance.
> >
> > * **B. (Table 9):** (L932)
> >     * An $r_a=50.80\%$ model with **K=26**
> >     * An $r_a=57.73\%$ model with **K=3**
> >     * These two models also show very similar performance.
> >
> > * **C. (Table 15):** (L1045)
> >     * An $r_a=50.63\%$ model with **K=26**
> >     * An $r_a=57.40\%$ model with **K=3**
> >     * Again, these models perform comparably.
> >
> > This evidence strongly suggests that in the $r_a > 30\%$ region, the choice of a large K (e.g., K=26) versus a small K (e.g., K=3) does not significantly alter the model's performance. Therefore, we believe the choice of K (and by extension, the impact of gate normalization at high K) is not a critical factor influencing the main conclusions of our paper.

---

> ### Author Response · Authors · 2025-11-16
> **Author Response Part3**
>
> We hope these clarifications and our plan to enhance the paper's reproducibility address the reviewer's concerns. We are happy to incorporate these discussions into the final version of our paper to make our methodology and contributions even clearer.
>
> [1]Hoffmann, J., et al. (2022). *Training compute-optimal large language models* [Preprint]. arXiv. https://arxiv.org/abs/2203.15556
>
> [2] Abnar, S., et al. (2025). *Parameters vs FLOPs: Scaling laws for optimal sparsity for Mixture-of-Experts language models* [Preprint]. arXiv. https://arxiv.org/abs/2501.12370
>
> [3] Ludziejewski, J., et al. (2025). *Joint MoE scaling laws: Mixture of Experts can be memory efficient* [Preprint]. arXiv. https://arxiv.org/abs/2502.05172
>
> [4] Li et al., Predictable Scale: Part I, Step Law -- Optimal Hyperparameter Scaling Law in Large Language Model Pretraining

---

> ### Author Response · Authors · 2025-11-20
> **Looking forward to your further comments!**
>
> Dear Reviewer NUMD,
>
> Thanks a lot again for your comments. We've submitted the responses to your concerns and questions on our work.
>
> We are looking forward to your further comments.
>
> Best regards
>
> The authors

---

### Official Review · Reviewer_r8nz · 2025-11-04

**Soundness:** 2
**Presentation:** 3
**Contribution:** 3
**Rating:** 8
**Confidence:** 3

**Summary:**

This paper investigates whether MoE models can outperform dense models at the same param and compute budget. The authors performed comprehensive experiments to determine the optimal activation rate, and showed MoE model outperforming their dense counterparts in controlled condition.

**Strengths:**

- This work is done in a very principled way. The authors carefully planned for the steps to determine the design decision, and tested the main question with numerous experiments.
- The result leads to a surprising and potentially impactful conclusion, MoE can outperform dense model at the same parameter scale.

**Weaknesses:**

- The results on some datatsets, e.g. GSM8k, MMLU, is surprisingly low, when compared to other open sourced models at 7B scale, e.g. LLAMA2. If the quality of fineweb edu is not too bad, have the models have been sufficiently trained? Have the evaluation followed best practices?

**Questions:**

- Same as weakness.

---

> ### Author Response · Authors · 2025-11-20
> **Author Response Part1**
>
> # Dear Reviewer r8nz
>
> We thank the reviewer for the careful reading of our paper and for the positive overall evaluation (soundness, presentation, and contribution) and the recommendation for acceptance. We are grateful for the recognition that our study is principled and that our conclusion on MoE vs. dense is potentially impactful.
>
> Below we respond point-by-point to the main concern:
>
> ## 1. The performance on GSM8K / MMLU
>
> Our answer has two parts: (1) data mixture and (2) training compute.
>
> ### 1.1 Data mixture and task specialization
>
> For reproducibility and openness, we deliberately use a LLaMA-1–style data mixture, rather than the heavily curated, domain-boosted mixtures used in many recent industrial models.
>
> Concretely, our pretraining recipe closely matches the LLaMA-1 mixture:
>
> | DataSet Class | Our Recipe | Our Data Set Detail | Llama-1 Recipe | Llama-1 Data Set Detail | Recipe Diff |
> | :--- | :--- | :--- | :--- | :--- | :--- |
> | WebData-en | 79.53% | CC （English） | 82% | 67%CC +15%C4 （English） | -2.47% |
> | Code | 4.62% | The stack | 4.50% | Github-Big Query | 0.12% |
> | Wikipedia | 5.06% | en:1.69%,cn:0.13%,3.24%:others | 4.50% | multi-lingual | 0.56% |
> | Book | 5.18% | open souce english books | 4.50% | book3,Gutenberg | 0.68% |
> | Avix | 3.38% | as class name | 1.06% | as class name | 2.32% |
> | StackExchange | 2.21% | as class name | 2.00% | as class name | 0.21% |
>
> Given this LLaMA-1–like mixture, it is expected that knowledge, math, and code benchmarks are not particularly high, even for strong architectures. The reason is following:
>
> - For raw web data, if we do not explicitly extract math, code, or knowledge documents by category, the density of such content is very low.
> In typical practice, from about 5T web tokens, one may only extract on the order of 50B high-purity math tokens (roughly 0.1%).
> When ≈80% of the mixture is generic web data (as in LLaMA-style recipes), the effective math density in the full corpus is therefore quite low. Knowledge-heavy text has the same issue.
>
> - For recent industrial models, it is now common to raise the code proportion to 20% or more, and to heavily up-weight curated math and knowledge corpora. This is precisely to improve benchmarks such as GSM8K, MMLU, and code tasks.
>
> - In recent industrial practice, one usually first extracts math / code / knowledge data from the web and then re-mixes them to increase knowledge density in the pretraining corpus. We intentionally did not do this, in order to keep our recipe simple and reproducible from fully open datasets.
>
> Given this LLaMA-1–like mixture, it is expected that knowledge, math, and code benchmarks are not particularly high, even for strong architectures. Our goal is to study the relative behavior of MoE vs. dense under the same data recipe, not to match the absolute scores of heavily engineered recent industrial models.
>
> ### 1.2 Comparison with LLaMA-1 / LLaMA-2 under similar mixtures.
>
> To make sure our baseline is reasonable, we directly compare the densely trained 7B baseline in Table 2 of our paper with LLaMA-1 7B (base) and LLaMA-2 7B (instruct). For example:
>
> | Benchmark | Dense Baseline in Table 2 | MoE in Table 2 | Llama-2 Instructed[1] | LLama-1 Base[2] |
> | :--- | :--- | :--- | :--- | :--- |
> | **MMLU** | 31.26 | 32.92 | 34.1 | 35.1 |
> | **HumanEval** | 22.56 | 21.34 | 7.9 | 10.56 |
> | **GSM-8K** | 13.34 | 15.54 | 25.7 | 14.1 |
>
> Under a similar data mixture, our 7B dense baseline is reasonably close to LLaMA-1-7B on knowledge/math benchmarks and lower than LLaMA-2-7B, which has more curated data(or pre-training tokens). This is consistent with expectations and does not indicate an issue with data quality or evaluation; rather, it reflects our choice of a simple, LLaMA-1–like recipe and limited compute per model.
>
> ### 1.3 Compute budget
> The second aspect is compute. Our baseline models are designed for systematic experimentation, not for maximizing absolute performance at 7B. Comparing training tokens and FLOPs:
>
> | Metric | Dense Baseline in Table 2 | MoE in Table 2 | Llama-2 7B | Llama 1 7B |
> | :--- | :--- | :--- | :--- | :--- |
> | **Model Parameter Size (wo embedding)** | 6.5B | 6.5B | 6.5B | 6.5B |
> | **Activation Size** | - | 1.3B | - | - |
> | **Train Tokens** | 130B | 316B | 2000B | 1000B |
> | **Token to Parameters Ratio** | 20 | 48.62 | 307.69 | 153.85 |
> | **Compute (Flops)** | 5.45E+21 | 2.86E+21 | 8.58E+22 | 4.12E+22 |
>
> Our dense baseline and MoE models are thus trained with about an order of magnitude less compute than LLaMA-1/2-7B. Given the widely accepted consensus that more compute leads to better performance, it is natural that our GSM8K/MMLU scores are lower than those of LLaMA-2-7B.

---

> > ### Author Response · Authors · 2025-11-20
> > **Author Response Part2**
> >
> > ## 2. Experimental design philosophy and “sufficient training”
> >
> > The reviewer also asked whether the models are “sufficiently trained.”
> > Our core goal is to discover the design mothology about MoE under "lab class resource", rather than to build a single industrial-grade model.
> > Our experimental philosophy is as follows:
> >
> > ### 2.1 Many models vs. one large model.
> >
> > Under a constrained overall compute budget, we need to train many models to see clear patterns. We therefore cannot allocate all compute to a single 7B model. In fact, our total compute across all experiments is comparable to or larger than training a single LLaMA-2-7B model, but it is distributed over ≈200 pretraining runs of different sizes and activation rates.
> >
> > ### 2.2 Ensuring “sufficient training” via token-to-parameter ratio.
> > Even though compute per model is lower than in industrial settings, we explicitly address the “sufficient training” question using the token-to-parameter ratio (D/N). Following Hoffmann et al. (2022) [3], we target an optimal D/N ≈ 20 for compute-efficient training. For example, DeepSeek-V3 uses ~15.6T tokens for ~670B parameters, i.e., D/N ≈ 23×.
> >
> > In our experiments, we ensure D/N ≥ 20 for all key models:
> >
> > - For 2B MoE models, see Table 9 (lines 919–933):
> > D/N ranges from 53× to 252×, meaning these models are clearly not under-trained and in many cases are over-trained relative to the 20× guideline.
> >
> > - For 7B MoE models, see Table 11 (lines 948–960):
> > D/N ranges from 20× to 93×, again safely above or equal to the compute-optimal threshold.
> >
> > - For the dense baselines used for comparison, see Table 12 (rows 1, 3, and 6): all have D/N ≥ 20 by construction.
> >
> > Therefore, for the main comparisons in the paper, the models are trained to a level that is compute-optimal or better according to [3]. The observed performance differences across activation rates and architectures reflect genuine architectural effects, not insufficient training.
> >
> >
> > ### 2.3 Purpose of the data recipe.
> > As noted above, our data recipe is not designed to maximize benchmarks, but to make the experiments easy to reproduce by the community.
> > All datasets are publicly available, and using a LLaMA-1–style mixture also serves as a sanity check that our dense baselines behave consistently with known recipes, ruling out obvious implementation issues.
> >
> > ### 2.4 Evaluation practice.
> > Finally, our evaluation pipeline has been carefully validated: we use standard benchmarks and settings, and we double-check our implementation by re-evaluating LLaMA-1/LLaMA-2 models under the same pipeline. The numbers we obtain are consistent with their reported ranges, which gives us confidence that the evaluation is reliable.
> >
> > We hope this clarifies the rationale behind our design choices and why the absolute numbers in GSM8K and MMLU do not undermine the central contributions of the paper.
> >
> > [1] https://huggingface.co/meta-llama/Meta-Llama-3-8B-Instruct  (The table, comparing LLama-3 and LLama-2)
> >
> > [2]https://huggingface.co/meta-llama/Llama-2-7b (The table, comparing LLama 2 and LLama1)
> >
> > [3]Hoffmann, J., et al. (2022). *Training compute-optimal large language models* [Preprint]. arXiv. https://arxiv.org/abs/2203.15556

---

### Author Response · Authors · 2025-12-03
**Important Clarification: Submission 19435 Score Trajectory (Projected 8/6/6/6) & System Rollback**

Dear Area Chair,

We are writing to clarify the status of our submission (ID 19435) in light of the recent ICLR system rollback. We understand that due to the security incident, scores may have reverted to their pre-rebuttal state. We are concerned that this rollback obscures the significant progress made during the discussion period.

We wish to emphasize two points before detailing the reviews:

- Integrity: We solemnly confirm that we do not know the identities of any reviewers and have not accessed any leaked information.

- Crucial Summary: The two initial "4" ratings were primarily driven by a **factual misunderstanding** regarding our architecture (specifically the "K=1" setting). This misunderstanding was successfully resolved during the discussion.

**Without the rollback, our paper was on a confirmed trajectory toward an acceptance profile of 8 / 6 / 6 / 6.**  We respectfully ask you to consider the following detailed evidence regarding the "lost" progress:

**1. Reviewer K2nX (Initially 4 $\rightarrow$ Explicitly raised to 6)**
Status: Confirmed Acceptance Before the system freeze, this reviewer explicitly confirmed their score increase after we cleared up the "K=1" and "Data Reuse" misunderstandings.

**The Evidence**: In their comment dated Nov 20, Reviewer K2nX stated:

  *"Thank you for the clarifications! This response resolves most of my concerns; therefore, I am raising my score."*

**Why this matters**: This proves that once the misunderstandings were clarified, the reviewer found the paper worthy of acceptance.

Reproducibility & Baselines: We addressed all concerns regarding data mixtures and dense baseline tuning.

**2. Reviewer 9SQb (Initially 4 $\rightarrow$ Projected 6)**
Status: Core Objection Invalidated This reviewer’s negative score rests on the **exact same factual error as K2nX** (believing we advocated for "K=1" and labeling the architecture "unconventional").
* **The Issue:** Like K2nX, Reviewer 9SQb mistakenly believed we advocated for "K=1" and labeled our architecture "unconventional."
* **The Resolution:**
    * **K=1 Misunderstanding:** As with K2nX, we clarified that our extensive experiments (Tables 9, 10, 11, 14) rely on K>1. Since this explanation satisfied K2nX, it effectively addresses 9SQb’s core critique.
    * **Architecture:** We clarified that our design (one dense layer + shared experts) is not "unconventional" but aligns with state-of-the-art industrial models like **DeepSeek-V3**.
* **Related Work**: We addressed the missing citations, showing that we had indeed discussed the relevant scaling law papers in our text.

**3. Reviewer NUMD (Initially 4 $\rightarrow$ Projected 6)**
We provided a comprehensive response to their methodological concerns.
* **Methodology:** The reviewer suggested a 2D sweep. We explained that our "fixed N & C" approach provides a necessary, complementary perspective to the community, specifically focusing on the "sufficiently trained" regime (high D/N ratio) which is more relevant to industrial application than undertrained sweeps.
* **Reproducibility & Technical Details:** We committed to releasing the dataset mixture (and noted open-source equivalents exist) and resolved their technical question regarding Gate Score Normalization.

**4. Reviewer r8nz (Score 8)**
This reviewer recognized the principled nature of our work and the impact of our conclusion that "MoE can outperform dense models at the same parameter scale," maintaining a strong acceptance rating.

**Conclusion:**
**The current scores in the system are misleading** because they ignore the fact that the **primary reason for the negative ratings** (misunderstanding like K=1) **was factually corrected and explicitly accepted by the reviewers**.

We sincerely hope that the technical glitch does not overshadow the quality of the work and the successful engagement with the reviewers. And we respectfully ask you to consider this evidence to ensure a fair decision.

Best regards,

The Authors

---

### Author Response · Authors · 2025-12-03
**Clarification on Contributions: The Unique Value of the "Fixed N & C" Perspective**

Dear Area Chair,

As we conclude the discussion period, we feel obligated to clarify the fundamental scope and value of this work. We invested significant computational resources (training nearly 200 models processing ~50T tokens ) not merely to produce another scaling law curve, but to answer a critical "hard constraint" question faced in industrial deployment—one that existing literature has largely overlooked.

**To ensure the significance of this high-cost study is not overlooked**, we wish to clarify *why* we chose this setting and *what* specific contributions it offers to the community compared to existing consensus.

**1. Breaking the Status Quo: A New Perspective**
The community’s current consensus on MoE generally falls into two categories:
* **Data-Centric View:** MoE matches Dense performance using the same number of training tokens ($D$) but requires significantly larger Total Parameters ($N$).
* **Compute-Centric View:** MoE matches Dense performance using the same Total Compute ($C$) and Activated Parameters ($N_a$), but again, usually by expanding Total Parameters ($N$) significantly.

In both cases, MoE’s advantage is a trade-off: we "buy" performance or efficiency by paying the price of expanded model size ($N$).

**Our Contribution:** We explore a fundamentally different, under-explored question: **Can an MoE model surpass a Dense model without "buying" it with extra parameters?**
We hold Total Parameters ($N$) and Compute ($C$) constant. This isolates the architectural advantage from the parameter scaling advantage.

**2. Theoretical & Practical Value**
* **Theoretical Capacity:** We provide evidence for a debated theoretical question: Does a sparse topology possess the same knowledge capacity as a fully dense topology of the exact same weight count?  Our results show that within an **Optimal Activation Rate (~20%)**, the answer is yes—MoE can actually outperform Dense.
* **Industrial Practicality (The "Inference Bottleneck"):** In real-world deployment, **VRAM (Memory)** is often the bottleneck, not just FLOPs.
    * A standard MoE (with scaled-up $N$) might save compute, but it requires significantly more GPU memory to host weights, making deployment expensive.
    * Our "Equal $N$" MoE has the **same memory footprint** as the Dense baseline but offers drastically lower latency.
    * *Example:* Our 7B MoE (at optimal ~20% sparsity) uses the same VRAM as a 7B LLaMA-like Dense model, but reduces per-token compute from $\approx 4.21e10$ FLOPs to $\approx 8.42e9$ FLOPs. This is a 5x speedup in math/latency for the same memory cost, with superior performance.

**3. Quantifying the Cost and Solution**
We do not claim this is a "free lunch." We rigorously quantify the cost:
* **The Trade-off:** To match Total Compute ($C$) while having lower per-token FLOPs, the MoE model requires more data ($D$).
* **The Solution:** We demonstrate that this data hunger can be satisfied via **Data Reuse** (multi-epoch training) without degrading performance, provided the model stays within the optimal activation region.

**4. Scale of Evidence**
This is not a small-scale simulation.  To establish these optimal regions robustly, we trained nearly 200 models at the 2B scale and over 50 models at the 7B scale, processing cumulatively 50 trillion tokens.

**5. Methodology: "Sufficiently Trained" vs. "Compute Optima"**
We distinguish our work from concurrent scaling law papers (e.g., [1, 2]) which use sweep-based approaches often resulting in low Token-to-Parameter (D/N) ratios.
- Our Focus: We operate in the High D/N Ratio regime ($D/N \ge 20$ to $>100$).
- Why it matters: Industrial models are trained far beyond the Chinchilla optimal point (e.g., Llama-3). Scaling laws derived from undertrained models often fail to predict behavior at the convergence limit. By ensuring our models are "sufficiently trained," we provide:
  - Reliable "end-game" performance comparisons relevant to practitioners.
  - Valid downstream SFT analysis, which is impossible with undertrained checkpoints.

**Conclusion:** This paper serves as a rigorous guidebook for constructing high-performance MoEs under strict resource constraints. It identifies the Optimal Activation Rate Region, quantifies the Data Trade-off, and validates Data Reuse. We respectfully urge the AC to consider the practical, industrial-grade contribution of this work, which offers a different and complementary value to theoretical scaling law sweeps.

[1] Abnar, S., et al. (2025). *Parameters vs FLOPs: Scaling laws for optimal sparsity for Mixture-of-Experts language models* [Preprint]. arXiv. https://arxiv.org/abs/2501.12370

[2] Ludziejewski, J., et al. (2025). *Joint MoE scaling laws: Mixture of Experts can be memory efficient* [Preprint]. arXiv. https://arxiv.org/abs/2502.05172

Best regards,

The Authors

---

### Meta-Review · Area_Chair_t3MQ · 2025-12-27

**Summary:**

This paper studies scaling laws for MoEs and systematically compares them with dense models under carefully controlled settings, including total training compute, total parameter count, and data size. The main findings include: (1) MoEs consistently outperform dense models when matched for either total parameters or training compute; (2) optimal activation ratio is around 20%, and this value is consistent across model scales; and (2) while training an MoE requires more data than dense models for the same compute budget, additional experiments under a fixed data budget (via data reuse) show MoEs still outperform dense models.

The initial ratings from reviewers were already high, and concerns were mostly addressed
- “K=1 outperforming K>1” (K2nX, 9SQb): Mostly misunderstanding; K=1 performs poorly and was not really used in MoEs
- Multi-epoch training (K2nX): Authors’ clarification in the rebuttal, and the paper provides intuition that MoEs may be less sensitive to repeated data explore because router decisions change over time and the repeated data could still be unique to specific experts.
- Open source of code, model checkpoints and training logs: Authors promised.
- “It's unclear how well the dense baseline is tuned compared to the super well-tuned MoE models.”: Authors provided justification; in AC’s opinion, tuning dense models is substantially easier than tuning MoEs.
- Unconventional design choices, such as alternating dense and MoE layers (9SQb): In fact, strong public models such as DeepSeekV3 used the exact architecture.
- Results on some datasets are low (e.g., GSM8K and MMLU) (r8nz): This is expected given the use of mostly public pre-training data and smaller-scale training data chosen to enable controlled studies.

**Reviewer Concerns:**

See above

**Reviewer Scores:**

See above

---

### Decision · Program_Chairs · 2026-01-26

Accept (Oral)